# Revisiting Generative Policies: A Simpler Reinforcement Learning Algorithmic Perspective

## Abstract

Generative models, particularly diffusion models, have achieved remarkable success in density estimation for multimodal data, drawing significant interest from the reinforcement learning (RL) community, especially in policy modeling in continuous action spaces. However, existing works exhibit significant variations in training schemes and RL optimization objectives, and some methods are only applicable to diffusion models. In this study, we compare and analyze various generative policy training and deployment techniques, identifying and validating effective designs for generative policy algorithms. Specifically, we revisit existing training objectives and classify them into two categories, each linked to a simpler approach. The first approach, Generative Model Policy Optimization (GMPO), employs a native advantage-weighted regression formulation as the training objective, which is significantly simpler than previous methods. The second approach, Generative Model Policy Gradient (GMPG), offers a numerically stable implementation of the native policy gradient method. We introduce a standardized experimental framework named *GenerativeRL*. Our experiments demonstrate that the proposed methods achieve state-of-the-art performance on various offline-RL datasets, offering a unified and practical guideline for training and deploying generative policies.

## 1 Introduction

Generative models, such as flow models and diffusion models, have demonstrated remarkable capabilities in modeling multi-modal data across diverse applications, including image, video, and audio generation (Rombach et al., 2022; Ho et al., 2022; Mittal et al., 2021), and protein structure prediction (Abramson et al., 2024). Their expressive power stems from constructing continuous and invertible mappings between probability distributions, enabling the transformation of simple distributions like standard Gaussians into complex target distributions. Generative policies, which are RL policy models based on generative models, have become a focus of study within the RL community (Janner et al., 2022; Chi et al., 2023; Ren et al., 2024). They offer a principled approach to modeling expressive and nuanced action distributions, particularly important in robotics tasks with continuous and high-dimensional action spaces.

Despite the success of recent studies in offline-RL (Chen et al., 2023; Wang et al., 2023; Lu et al., 2023; Hansen-Estruch et al., 2023), these studies often employ complex training schemes and lack systematic investigation. This hinders understanding of key factors in training generative models for policy modeling, leading to unnecessary dependencies, training inefficiencies, and higher inference costs. Some work focuses primarily on diffusion models (Sohl-Dickstein et al., 2015; Ho et al., 2020), which has limited applicability to other generative models, such as flow models (Lipman et al., 2023; Liu et al., 2023; Pooladian et al., 2023; Albergo & Vanden-Eijnden, 2023; Tong et al., 2024).

This motivates us to develop simple yet effective training schemes for both diffusion and other emerging generative models, leveraging their advancements for the RL community's benefit. We revisit previous works on generative policy optimization in Table 1, and then categorize these works, and propose two training schemes: Generative Model Policy Optimization (GMPO) and Generative Model Policy Gradient (GMPG). GMPO is an advantage-weighted regression method with a stable training process that does not require pretraining the generative model before optimal policy extraction. This makes it more efficient and easier to train, featuring a shorter training schedule and a wider range of model applications while maintaining comparable performance to previous works. GMPG

is a policy gradient-based method in an RL-native formulation. We provide a numerically stable implementation for continuous-time generative models as an RL policy. It has proven effective in optimal policy extraction, especially for suboptimal policies. We evaluate the performance of diffusion and flow models using these two training schemes in an offline reinforcement learning setting with the D4RL dataset (Fu et al., 2021) and RL Unplugged (Gulcehre et al., 2020). Our results demonstrate that the proposed schemes offer comparable or better performance than previous works in most cases. We provide multiple ablations, covering model types, temperature settings, solver schemes, and sampling time steps for training, to validate the effectiveness of our proposed methods.

To facilitate consistent comparisons and analyses, we introduce a standardized experimental framework designed to combine the strengths of generative models and reinforcement learning by decoupling the generative model from the RL components. This decoupling enables consistent comparisons and analyses of different generative models within the same RL context, a capability lacking in previous works. Four key properties distinguish our framework from existing ones: unified API, flexible data formats and shapes, auto-grad support, and diverse generative model integration.

In conclusion, the main contributions of our work include:

- Proposing two simple yet effective RL-native training approaches for generative policies, GMPO and GMPG, which are compatible for both diffusion and flow models.

- Evaluating the performance of different generative models with the two training schemes on offline-RL dataset, achieving state-of-the-art performance in most cases. Our work elucidates the key factors in training generative policies and corrects several biases described in previous research.

- Providing a unified framework for combining the power of generative models and RL seamlessly, easy for conducting a fair and RL-native evaluations, named *GenerativeRL*.

## 2   RELATED WORKS

**Generative Policy Algorithms.**   Haarnoja et al. (2017) introduces Soft Q-Learning and uses an energy-based generative policy for learning continuous actions from continuous states. Haarnoja et al. (2018a) integrates hierarchical policies with discrete layers into the Soft Actor-Critic (Haarnoja et al., 2018b) framework, which is called SACLSP and utilizes normalizing flows. Janner et al. (2022) first incorporate diffusion models into RL named as diffuser, acting as optimal trajectory planners by linking generative models with value function guidance being applied rudimentary. Chi et al. (2023) explores diffusion models for policy modeling in robotics, which is the concept of diffusion policy is first introduced. Chen et al. (2023) demonstrates effective policy learning in offline Q-learning using a diffusion model as support called SfBC. Wang et al. (2023) introduces Diffusion-QL and achieves policy regularization by alternating training between a Q-function-guided diffusion model and a diffusion-supported Q-function. Lu et al. (2023) introduces Q-guided Policy Optimization (QGPO) and derives the exact formulation of energy guidance for energy-conditioned diffusion models, enabling precise Q-guidance for optimal policy. Hansen-Estruch et al. (2023) introduces Implicit Diffusion Q-learning (IDQL) which uncovers the implicit policy form after Implicit Q-learning, emphasizing the importance of sampling from both the behavior policy and the diffusion model. Chen et al. (2024) uses the score function of a pre-trained diffusion model as a regularizer, optimizing a Gaussian policy to maximize the Q-function by combining the multimodal properties of diffusion models with the fast inference of Gaussian policies, which is called Score Regularized Policy Optimization (SRPO). Similar methods can also be applied to models trained using the flow matching (Zheng et al., 2023b; Kim et al., 2024). More details about the generative policy analyzed in this paper are provided in Appendix C.

**Reinforcement Learning Frameworks.**   Several open-source RL frameworks provide unified interfaces for solving RL problems, although some are no longer maintained, such as OpenAI Baselines (Dhariwal et al., 2017), Facebook/ELF (Tian et al., 2017), TFAgents (Guadarrama et al., 2018), and JaxRL (Kostrikov, 2021). Active and widely-used frameworks include RLlib (Liang et al., 2018), which focuses on distributed RL for training large-scale models; Dopamine (Castro et al., 2018), a research framework for rapid RL algorithm prototyping; and Acme (Hoffman et al., 2020), designed for flexibility and scalability in RL research. Stable Baselines3 (Raffin et al., 2021), Tianshou (Weng et al., 2022), DI-engine (Niu et al., 2021), ElegantRL (Liu et al., 2021), and

CleanRL (Huang et al., 2022) offer a wide range of algorithms, environments, and user-friendly interfaces. TorchRL (Bou et al., 2023), a modular RL framework, supports PyTorch's native API and excels in handling dictionary-type tensors common in RL. The framework most related to ours is CleanDiffuser (Dong et al., 2024), which was recently proposed to integrate different types of diffusion algorithmic branches into a single framework.

## 3 BACKGROUND

This section provides a brief introduction to reinforcement learning and generative models. For more fundamentals of diffusion and flow models, please refer to Appendix B.

### 3.1 REINFORCEMENT LEARNING

Reinforcement learning (RL) addresses sequential decision-making tasks typically modeled as a Markov decision process (MDP), defined by the tuple $(\mathcal{S}, \mathcal{A}, p, r, \gamma)$. Here, $\mathcal{S}$ is the state space, $\mathcal{A}$ is the action space, $p(s_{t+1}|s_t, a_t)$ represents the transition dynamics, $r(s_t, a_t)$ is the reward function, and $\gamma$ is the discount factor. At each time step $t$, the agent, in state $s_t$, takes an action $a_t \in \mathcal{A}$ according to the policy $\pi(a_t|s_t)$. The agent then receives a reward $r_t$ and transitions to a new state $s_{t+1}$ based on the transition dynamics $p(s_{t+1}|s_t, a_t)$. The goal of the agent is to learn a policy $\pi : \mathcal{S} \to \mathcal{A}$ that maximizes the expected cumulative reward over time, using only previously collected data. This objective can be expressed as: $\mathcal{R}_{s_0, a_0} = \mathbb{E}_{s_0, a_0, s_1, a_1, \ldots}\left[\sum_{t=0}^{\infty} \gamma^t r_t\right]$.

In offline reinforcement learning (offline-RL), the agent has access to a fixed dataset $\mathcal{D}_\mu$ of historical interaction trajectories $\{s_t, a_t, r_t, s_{t+1}\}$, collected by a behavior policy $\mu(a_t|s_t)$, which is often suboptimal. Offline-RL is challenging because the agent cannot collect new data to correct its mistakes, unlike in online-RL, where exploration is possible.

Suppose the value of action $a$ at state $s$ is modeled by a Q-function $Q(s, a) \approx \mathcal{R}_{s,a}$, which estimates the expected return of taking action $a$ at state $s$ and following policy $\pi$ thereafter. The value of state $s$ is modeled by a V-function $V(s) = \mathbb{E}_{a \sim \pi(\cdot|s)}[Q(s, a)]$.

### 3.2 GENERATIVE MODELS

**Diffusion Models.** Given a fixed data or target distribution, a diffusion model is determined by the diffusion process path as an stochastic differential equation (SDE): $\mathrm{d}x = f(t)x_t\mathrm{d}t + g(t)\mathrm{d}w_t$. The transition distribution of a point $x \in \mathbb{R}^d$ from time 0 to $t$ is: $p(x_t|x_0) \sim \mathcal{N}(x_t|\alpha_t x_0, \sigma_t^2 I)$. The reverse process path can be described by an ordinary differential equation (ODE):

$$\frac{\mathrm{d}x_t}{\mathrm{d}t} = v(x_t) = f(t)x_t - \frac{1}{2}g^2(t)\nabla_{x_t} \log p(x_t), \tag{1}$$

where $v(x_t)$ is the velocity function and $\nabla_{x_t} \log p(x_t)$ is the score function, typically modeled by a neural network with parameters $\theta$, denoted as $v_\theta(x_t)$ and $s_\theta(x_t)$, respectively.

**Flow Models.** Consider a flow model with time-varying velocity $v(x_t)$, whose flow path can be described by an ODE: $\frac{\mathrm{d}x_t}{\mathrm{d}t} = v(x_t)$. The velocity field transforms the source distribution $p(x_0)$ at time $t = 0$ into the target distribution $p(x_1)$ at time $t = 1$ along the flow path and conforms to the continuity equation: $\frac{\partial p}{\partial t} + \nabla_x \cdot (pv) = 0$.

**Model Training.** Training continuous-time generative model involves a matching objective $\mathcal{L}_{\text{Matching}}(\theta)$, including the Score Matching method (Hyvärinen, 2005; Vincent, 2011):

$$\mathcal{L}_{\text{DSM}} = \frac{1}{2}\int_0^1 \mathbb{E}_{p(x_t, x_0)}\left[\lambda(t)\|s_\theta(x_t) - \nabla_{x_t}\log p(x_t|x_0)\|^2\right]\mathrm{d}t, \tag{2}$$

and the Flow Matching method (Lipman et al., 2023; Tong et al., 2024):

$$\mathcal{L}_{\text{CFM}} = \frac{1}{2}\int_0^1 \mathbb{E}_{p(x_t, x_0, x_1)}\left[\|v_\theta(x_t) - v(x_t|x_0, x_1)\|^2\right]\mathrm{d}t. \tag{3}$$

Table 1: Training schemes of different generative policy RL algorithms. The "Suitable Generative Model" column indicates the model types to which the algorithm can be applied, with parentheses showing the models actually used in previous work. If the model type is "Any," this method is applicable to all generative models, including diffusion and flow models discussed in this paper. The "Behavior Policy" column indicates whether the algorithm requires a pre-trained policy model for subsequent training. The "Critic Training" column describes the scheme used to learn the Q-function. The "Optimal Policy Extraction" column indicates whether the method trains and extracts the optimal policy.

| Algorithm | Suitable Generative Model | Behavior Policy | Critic Training | Optimal Policy Extraction |
|---|---|---|---|---|
| SfBC | Any (VPSDE) | Needed | In-support Q-Learning | ✗ |
| QGPO | Diffusion (VPSDE) | Needed | In-support Q-Learning | ✓ |
| Diffusion-QL | Any (DDPM) | Needed | Conventional Q-Learning | ✓ |
| IDQL | Any (DDPM) | Needed | IQL | ✗ |
| SRPO | Diffusion (VPSDE) | Needed | IQL | ✓ |
| GMPO | Any | Not needed | IQL | ✓ |
| GMPG | Any | Needed | IQL | ✓ |

For diffusion models, we investigate the Linear Variance-Preserving SDE (VP-SDE) model proposed by Song et al. (2021b), a continuous-time variant of Denoising Diffusion Probabilistic Models (DDPM) by Ho et al. (2020), and the Generalized VP-SDE (GVP) model introduced by Albergo & Vanden-Eijnden (2023), which extends the Improved DDPM by Nichol & Dhariwal (2021) with triangular scale and noise levels. For flow models, we investigate a simple flow model named I-CFM by Tong et al. (2024).

## 4 REVISITING GENERATIVE POLICIES

We establish the formulation of the optimal policy in offline-RL in Section 4.1. Next, we review prior work on generative policies in Section 4.2, with additional details in Appendix C. Subsequently, Section 4.3 introduces two straightforward and effective training schemes.

### 4.1 OPTIMAL POLICY IN OFFLINE-RL

Previous works in offline-RL (Peters & Schaal, 2007; Peters et al., 2010; Abdolmaleki et al., 2018; Wu et al., 2020) formulate policy optimization as a constrained optimization problem. The policy is learned by maximizing the expected return, subject to the KL divergence constraint to the behavior policy: $\pi^* = \arg\max_\pi \mathbb{E}_{s\sim\mathcal{D},a\sim\pi(\cdot|s)}\left[Q(s,a) - \frac{1}{\beta}D_{\mathrm{KL}}(\pi(\cdot|s)\|\mu(\cdot|s))\right]$. This approach ensures the learned policy rarely acts outside the support of the behavior policy, thus avoiding extrapolation errors that could degrade performance, as emphasized by Kumar et al. (2019) and Fujimoto et al. (2019). The optimal policy has an analytical form, as shown by Peng et al. (2021): $\pi^*(a|s) = \frac{e^{\beta(Q(s,a)-V(s))}}{Z(s)}\mu(a|s)$, where $Z(s)$ is a normalizing factor, and $\beta$ is a temperature parameter. In practice, we can build a parameterized neural-net policy $\pi_\theta$ to approximate the optimal policy $\pi^*$. The policy model can be trained by minimizing the KL divergence:

$$\mathcal{L}(\theta) = \mathbb{E}_{s\sim\mathcal{D}}\left[D_{\mathrm{KL}}(\pi^*(\cdot|s)\|\pi_\theta(\cdot|s))\right] = \mathbb{E}_{s\sim\mathcal{D},a\sim\mu(\cdot|s)}\left[-\frac{e^{\beta(Q(s,a)-V(s))}}{Z(s)}\log\pi_\theta(a|s)\right] + C, \quad (4)$$

where $C$ is a constant that does not depend on $\theta$, or by using reverse KL divergence:

$$\mathcal{L}(\theta) = \mathbb{E}_{s\sim\mathcal{D}}\left[D_{\mathrm{KL}}(\pi_\theta(\cdot|s)\|\pi^*(\cdot|s))\right] = \mathbb{E}_{s\sim\mathcal{D},a\sim\pi_\theta(\cdot|s)}\left[-\beta Q(s,a) + D_{\mathrm{KL}}(\pi_\theta(\cdot|s)\|\mu(\cdot|s))\right] + C. \quad (5)$$

Comparison and derivation details of Eq. 4 and Eq. 5 are provided in Appendix A.

### 4.2 PREVIOUS WORKS ON GENERATIVE POLICY

Table 1 summarizes existing approaches for obtaining optimal generative policies. Table 2 provides an overview of the training and inference schemes of these algorithms.

Table 2: Generative model training and inference schemes of different generative RL algorithms. The forward KL method, which includes SfBC, QGPO, IDQL, and GMPO, incorporates an advantage-weighted regression term in its training or inference schemes. In contrast, the reverse KL method, including Diffusion-QL, SRPO, and GMPG, utilizes the Q-function for policy gradients with KL constraints. More comparisons are provided in Appendix A.2 and Appendix C.

| Algorithm | Training Schemes | | Inference Schemes |
|---|---|---|---|
| Forward KL | $\nabla_\theta \mathbb{E}_{s,a\sim\mu} \left[ -\frac{e^{\beta(Q(s,a)-V(s))}}{Z(s)} \log \pi_\theta(a|s) \right]$ | | |
| Reverse KL | $\nabla_\theta \mathbb{E}_{s,a\sim\pi_\theta} \left[ -\beta Q(s,a) + D_{\mathrm{KL}}(\pi_\theta(\cdot|s)||\mu(\cdot|s)) \right]$ | | |
| **Public Works** | | | |
| SfBC | $\nabla_\theta \mathcal{L}_{\mathrm{Matching}}(\theta)$ | (Eq. 2) | $a \sim \mathrm{softmax}_{a_i\sim\mu_\theta} Q(a_i,s)$ |
| QGPO | $\nabla_\phi \mathbb{E}_{t,s,a_i\sim\mu} \left[ -\frac{e^{\beta Q(s,a_i)}}{\sum_{i=1}^N e^{\beta Q(s,a_i)}} \log \frac{e^{\mathcal{E}_\phi(s,a_i,t)}}{\sum_{i=1}^N e^{\mathcal{E}_\phi(s,a_i,t)}} \right]$ | | $a \sim \pi, \nabla \log \pi = \nabla \log \mu + \nabla \log \mathcal{E}_\phi$ |
| Diffusion-QL | $\nabla_\theta \mathbb{E}_{s,a\sim\pi_\theta} \left[ -\beta Q(s,a) + \mathcal{L}_{\mathrm{Matching}}(\theta) \right]$ | | $a \sim \mathrm{softmax}_{a_i\sim\pi_\theta} Q(a_i,s)$ |
| IDQL | $\nabla_\theta \mathcal{L}_{\mathrm{Matching}}(\theta)$ | (Eq. 2) | $a \sim \frac{\left|\frac{\partial}{\partial V}(Q(s,a_i)-V(s))\right|}{|Q(s,a_i)-V(s)|}\mu_\theta(a_i|s)$ |
| SRPO | $\nabla_\theta \mathbb{E}_{t,s,a\sim\pi_\theta} \left[ -\beta Q(s,a) + w(t)(\epsilon_\psi(a_t|s) - \epsilon) \right]$ | | $a \sim \pi_\theta$ |
| **This work** | | | |
| GMPO | $\nabla_\theta \mathbb{E}_{s,a\sim\mu} \left[ \frac{e^{\beta(Q(s,a)-V(s))}}{Z(s)} \mathcal{L}_{\mathrm{Matching}}(\theta) \right]$ | | $a \sim \pi_\theta$ |
| GMPG | $\nabla_\theta \mathbb{E}_{s,a\sim\pi_\theta} \left[ -\beta Q(s,a) + \log \frac{\pi_\theta(a|s)}{\mu(a|s)} \right]$ | | $a \sim \pi_\theta$ |

- • KL divergence or its variant for behavior policy constraint.
- • Q function for policy gradient method.
- • Importance weight for advantage-weighted regression.

**Generative Model Type**    Table 1 highlights that QGPO and SRPO are limited to diffusion models, while other algorithms accommodate various generative models. QGPO's restriction arises from its reliance on the Contrastive Energy Prediction method (Eq. 30), which distills the Q function into an energy guidance model specifically designed for diffusion models (Appendix C.2). Similarly, SRPO's constraint stems from its score-regularized loss, which naturally aligns with diffusion models due to their inherent modeling of the score function. This compatibility, however, does not extend to non-diffusion models, such as flow models (see Appendix C.4 for further explanation).

Given the rapid advancements in generative modeling, we strive to establish a unified training scheme applicable to any generative model. By extracting effective designs from QGPO and SRPO algorithms, we can eliminate components that hinder generalization, training efficiency, and complexity without affecting performance

**Behavior Policy Pretraining and Critic Training.**    As shown in Table 1, all methods require pretraining a behavior policy. SfBC and IDQL rely on it for importance sampling during inference. QGPO leverages the behavior policy for sampling in conjunction with energy guidance (Eq. 31). SRPO utilizes a well-trained behavior policy to regularize the Gaussian model policy (Eq35). Furthermore, while QGPO and SfBC employ in-support Q-learning with data augmentation from the behavior policy for Q function training (Eq. 29), Diffusion-QL utilizes traditional Q-learning with a similar dependence on the behavior policy (Eq. 36).

All three methods use the behavior policy for data augmentation during Q function training, making it essential. In contrast, SRPO and IDQL use Implicit Q-Learning to train the Q function directly (Eq. 32), without needing a behavior policy. This decouples the training of the behavior policy and the Q function, allowing simultaneous training.

The slower sampling rates of generative models raise questions about the necessity of data augmentation in training process of SfBC, QGPO, and Diffusion-QL. Simpler approaches, such as the Implicit Q-Learning employed by SRPO and IDQL, could offer greater efficiency for Q function training

in generative policy optimization. This observation motivates us to investigate whether comparable performance can be achieved by directly leveraging the Q function learned by IQL to guide generative policy optimization. Specifically, we are interested in exploring the potential of explicitly extracting a generative policy from the IQL-trained Q function, a direction not explored in previous works.

**Optimal Policy Extraction.** As shown in Table 1, SfBC and IDQL do not perform explicit policy extraction. Instead, they use importance sampling to derive the optimal policy from the behavior policy by evaluating the Q function. Although parallel computation can save inference time, the total computational budget during inference scales with the number of actions sampled from the behavior policy. SRPO uses a Gaussian model to explicitly output the optimal policy for guidance distillation, reducing computational costs and speeding up deployment. However, Gaussian models are generally less expressive than generative models, which can limit their effectiveness (Chi et al., 2023; Ren et al., 2024). Therefore, explicit policy extraction using generative models is preferred for balancing computational efficiency and expressiveness.

### 4.3 GENERATIVE MODEL POLICY TRAINING

To improve upon previous methods, we propose two straightforward yet effective training schemes for generative model policy optimization, derived from Eq. 4 and Eq. 5, as shown in Table 2. These schemes satisfy three key requirements: (1) Generality: Ensure compatibility with any generative model through a simple and effective training process. (2) Decoupled Training: Train the Q-function directly using Implicit Q-Learning (IQL), minimizing reliance on generative sampling. (3) Explicit Policy Extraction: Consistently infer only one action at a time for model inference.

**Generative Model Policy Optimization.** Inspired by Song et al. (2021a), who demonstrated that training with a maximum likelihood objective is equivalent to score matching, we replace the log-likelihood term with the matching loss (Eq. 2) of the generative model. By keeping the exponential form of the advantage function as the importance weight, as in Eq. 4, we derive the following advantage-weighted regression training objective suitable for both diffusion and flow models:

$$\mathcal{L}_{\text{GMPO}}(\theta) = \mathbb{E}_{s \sim \mathcal{D}, a \sim \pi^*(\cdot|s)} \left[ \mathcal{L}_{\text{Matching}}(\theta) \right]$$
$$= \mathbb{E}_{s \sim \mathcal{D}, a \sim \mu(\cdot|s)} \left[ \frac{e^{\beta(Q(s,a)-V(s))}}{Z(s)} \mathcal{L}_{\text{Matching}}(\theta) \right]. \quad (6)$$

Although Eq. 6 looks similar to Eq. 4, with the log-likelihood term replaced by the matching loss, it can be derived independently. See Appendix C.6 for more details.

Unlike previous works, our GMPO approach does not have to use data augmentation from the behavior policy. This removes the necessity of behavior policy and uses only data from the offline dataset:

$$\mathcal{L}_{\text{GMPO}}(\theta) = \mathbb{E}_{(s,a) \sim \mathcal{D}_\mu} \left[ \frac{e^{\beta(Q(s,a)-V(s))}}{Z(s)} \mathcal{L}_{\text{Matching}}(\theta) \right]. \quad (7)$$

More details about GMPO are provided in Appendix C.6.

**Generative Model Policy Gradient.** This approach is directly derived from Eq. 5:

$$\mathcal{L}_{\text{GMPG}}(\theta) = \mathbb{E}_{s \sim \mathcal{D}} \left[ D_{\text{KL}}(\pi_\theta(\cdot|s) || \pi^*(\cdot|s)) \right]$$
$$= \mathbb{E}_{s \sim \mathcal{D}, a \sim \pi_\theta(\cdot|s)} \left[ -\beta Q(s,a) + D_{\text{KL}}(\pi_\theta(\cdot|s) || \mu(\cdot|s)) \right] \quad (8)$$
$$= \mathbb{E}_{s \sim \mathcal{D}, a \sim \pi_\theta(\cdot|s)} \left[ -\beta Q(s,a) + \log \pi_\theta(a|s) - \log \mu(a|s) \right].$$

As an RL-native policy gradient method, GMPG directly calculates the log-likelihood term. However, efficiently computing gradients for diffusion and flow models is challenging due to their forward sampling process, which involves solving an initial value problem within an ODE solver. To address this, we employ advanced techniques such as the adjoint method or Neural ODEs (as proposed by Chen et al. (2018)), ensuring computational feasibility. Additionally, we utilize the Hutchinson trace estimator (Hutchinson, 1990; Grathwohl et al., 2019) to compute the log-likelihood of the policy for continuous-time generative models. See Appendix C.7.1 for more details.

In contrast to GMPG's direct log-likelihood calculation, Wang et al. (2023) propose an alternative approach that replaces the log-likelihood term with a score matching loss (Eq. 37), and further mitigates computational costs by employing a low time step ($T = 5$). However, this method has limitations, as it may not scale effectively to high-dimensional spaces and could potentially compromise generation quality with such a low time step. More details about GMPG are provided in Appendix C.7.

We illustrate the intuitive sampling trajectories of GMPO and GMPG with a 2D toy example in Appendix C.8 to clarify how it works.

## 5 FRAMEWORK

Based on the research needs analyzed in previous sections, we create *GenerativeRL* for verifying and comparing various generative models and reinforcement learning algorithms. The key distinction of GenerativeRL is its standardized implementation and unified API for generative models, allowing researchers to access these models at the configuration level without dealing with complex details. When compared to existing frameworks like CleanDiffuser (Dong et al., 2024), *GenerativeRL* differs in its design principles:

- Unified API: It offers a simple API that maximizes compatibility with different kinds of generative models, avoiding immature algorithms.
- Flexible Data Formats: It ensures consistent data formats for long-term use, supporting inputs and outputs as PyTorch tensors, tensordicts (Bou et al., 2023), and treetensors (Contributors, 2021), given the prevalence of dict-type data in RL.
- Auto-Grad Support: Designed to support Neural ODEs, it facilitates gradient-based inference via ODE or SDE, which is useful for many RL policies.
- Diverse Model Integration: It integrates various generative models, including flow and bridge models, treating diffusion models as a special case.

Thus, *GenerativeRL* seamlessly incorporates both diffusion and flow models for various RL algorithms, while decoupling the generative model from RL components. This allows for consistent comparisons and analyses of different generative models within the same RL context. See usage examples in Appendix E.1 and framework structure in Appendix E.2.

## 6 EXPERIMENTS

### 6.1 EXPERIMENTAL SETUP

Experiments are conducted on the classical offline-RL environments, D4RL dataset (Fu et al., 2021) and RL Unplugged DeepMind Control Suite datasets (Gulcehre et al., 2020).

Following QGPO (Lu et al., 2023), we adopt the same U-Net architecture with three hidden layers for all generative policy algorithms. The sampling process is performed using the Euler-Maruyama method in an ODE solver with uniform time steps, $T = 1000$ in training for GMPG, and $T = 32$ in evaluation for both GMPO and GMPG. All evaluations are conducted and averaged over five random seeds. Performance scores on D4RL datasets are normalized as suggested by Fu et al. (2021). We re-implemented QGPO, IDQL and SRPO under the same experimental settings for fair comparisons, reporting both our implementation scores and the original scores from the respective papers.

More details about computation resources, hyperparameters, and training specifics can be found in Appendix D.1.

### 6.2 EXPERIMENTS AND ANALYSIS

We address two key questions: (1) Can simpler RL-native training schemes like GMPO and GMPG extract the optimal policy and achieve performance comparable to state-of-the-art algorithms on the classical offline-RL dataset? (2) What are the experimental differences between GMPO and GMPG, given that both forward KL and reverse KL theoretically point to the same optimal policy?

Table 3: Performance evaluation on D4RL datasets of different generative policies. We provide the original scores of SfBC, Diffusion-QL, QGPO, IDQL, and SRPO from their respective papers. SfBC and Diffusion-QL are not integrated into our framework due to the substantial modifications required. Other algorithms have been successfully implemented using our unified framework. A detailed comparison between the original scores and our implementation can be found in Appendix D, Table 9.

| Environment | SfBC | Diffusion-QL | QGPO | IDQL | SRPO | GMPO | GMPG |
|---|---|---|---|---|---|---|---|
| **Model type** | VPSDE | DDPM | VPSDE | VPSDE | VPSDE | GVP | VPSDE |
| **Function type** | $\epsilon(x_t, t)$ | $\epsilon(x_t, t)$ | $\epsilon(x_t, t)$ | $\epsilon(x_t, t)$ | $\epsilon(x_t, t)$ | $v(x_t, t)$ | $v(x_t, t)$ |
| **Pretrain scheme** | Eq. 2 | Eq. 2 | Eq. 2 | Eq. 2 | Eq. 2 | / | Eq. 3 |
| **Fintune scheme** | / | Eq. 37 | Eq. 30 | / | Eq. 35 | Eq. 40 | Eq. 8 |
| halfcheetah-medium-expert-v2 | 92.6 | 96.8 | $92.0 \pm 1.5$ | $91.7 \pm 2.4$ | $86.7 \pm 3.7$ | $91.9 \pm 3.2$ | $89.0 \pm 6.4$ |
| hopper-medium-expert-v2 | 108.6 | 111.1 | $107.0 \pm 0.9$ | $96.8 \pm 10.4$ | $100.8 \pm 9.3$ | $112.0 \pm 1.8$ | $107.8 \pm 1.9$ |
| walker2d-medium-expert-v2 | 109.8 | 110.1 | $107.3 \pm 1.3$ | $107.0 \pm 0.5$ | $118.7 \pm 1.4$ | $108.1 \pm 0.7$ | $112.8 \pm 1.2$ |
| halfcheetah-medium-v2 | 45.9 | 51.1 | $44.0 \pm 0.7$ | $43.7 \pm 2.8$ | $51.4 \pm 2.9$ | $49.9 \pm 2.7$ | $57.0 \pm 3.1$ |
| hopper-medium-v2 | 57.1 | 90.5 | $80.1 \pm 7.0$ | $72.1 \pm 17.6$ | $97.2 \pm 3.3$ | $74.6 \pm 21.2$ | $101.1 \pm 2.6$ |
| walker2d-medium-v2 | 77.9 | 87.0 | $82.8 \pm 2.7$ | $82.0 \pm 2.4$ | $85.6 \pm 2.1$ | $81.1 \pm 4.3$ | $91.9 \pm 0.9$ |
| halfcheetah-medium-replay-v2 | 37.1 | 47.8 | $42.5 \pm 1.7$ | $41.6 \pm 8.4$ | $47.2 \pm 4.5$ | $42.3 \pm 3.6$ | $50.5 \pm 2.7$ |
| hopper-medium-replay-v2 | 86.2 | 100.7 | $99.3 \pm 1.8$ | $89.1 \pm 3.1$ | $78.2 \pm 12.1$ | $97.8 \pm 3.8$ | $86.3 \pm 10.5$ |
| walker2d-medium-replay-v2 | 65.1 | 95.5 | $81.1 \pm 4.2$ | $80.4 \pm 9.2$ | $79.6 \pm 7.6$ | $86.4 \pm 1.7$ | $90.1 \pm 2.2$ |
| **Average (Locomotion)** | 75.6 | 88.0 | $81.8 \pm 2.4$ | $78.3 \pm 6.3$ | $82.8 \pm 5.1$ | $82.7 \pm 4.8$ | $87.3 \pm 3.5$ |

Table 3 shows the performance of various generative policy algorithms on D4RL environments, including SfBC (Chen et al., 2023), Diffusion-QL (Wang et al., 2023), QGPO (Lu et al., 2023), IDQL (Hansen-Estruch et al., 2023), and SRPO (Chen et al., 2024). Table 4 presents the performance of generative policies on the RL Unplugged DeepMind Control Suite datasets (Gulcehre et al., 2020). For reference, we include the performance of two classical offline RL algorithms: RABM (Siegel et al., 2020) and D4PG (Barth-Maron et al., 2018), the latter being the algorithm for which most of this dataset was collected. The average performance across these tasks demonstrates that GMPO and GMPG effectively solve challenging continuous control tasks, achieving competitive results compared with other state-of-the-art algorithms.

Hansen-Estruch et al. (2023) claims that using highly expressive models with importance weighted objectives can be problematic as such models can increase the likelihood of all training points regardless of their weight. And they find using advantage-weighted regression in the DDPM objective to not help performance, so they recommend sampling from behavior policy and filter out the high Q-value actions with the softmax importance sampling. However, our practice overturns the above claims and found that generative policy trained with advantage-weighted regression even from a scratch initialization can gain comparably equivalent performance to IDQL using resampling tricks, as shown in Table 3. Since utilizing the same Implicit Q-learning method, GMPO and GMPG both successfully extract optimal policies from the Q function. GMPO, a simpler variant of QGPO, achieves comparable performance on these datasets. This indicates that the advantage-weighted regression loss, a common component of both methods, is crucial for successful training.

Unlike Diffusion-QL, which separates the KL divergence into a simulation-free score matching loss but remains Q guidance through simulation, GMPG computes both Q guidance and the KL divergence directly through simulation, achieving similar performance. Despite using a significantly larger number of sampling steps ($T = 1000$) compared to Diffusion-QL ($T = 5$), GMPG does not exhibit computational difficulties in our implementation. This effectively addresses the limitations highlighted by Wang et al. (2023), where the maximum $T$ they could afford was 20, and they opted for $T = 5$ to balance performance and computational cost.

In addition, our experiments challenged the intuition that a complete 1000-step inference inherently leads to high computational costs, even with Neural ODEs. Our results demonstrate that with direct gradient guidance, no extra computational cost is required for optimization. For example, in halfcheetah-medium-v2-GMPG-GVP, optimal performance is achieved in 50-100 steps, taking 5-10 hours on an A100 GPU. Similarly, for halfcheetah-medium-v2-GMPO-GVP, optimal performance occurs at 240K-480K steps, also requiring 5-10 hours. Despite slower calculations for a 1000-step inference for one gradient step, it uses fewer training batches, resulting in similar overall time costs, ensuring no computation is wasted during training.

Table 4: Performance evaluation on RL Unplugged DeepMind Control Suite dataset of different generative policies. We evaluate different generative policies using the RL Unplugged DeepMind Control Suite dataset. D4PG is the primary algorithm used for most of this dataset's collection, while RABM serves as a classical offline-RL algorithm for comparison. We present the original scores of D4PG and RABM from their respective papers. Since the original papers for QGPO, IDQL, and SRPO do not provide performance metrics on this dataset, we report the scores from our implementations.

| Environment | D4PG | RABM | QGPO | IDQL | SRPO | GMPO | GMPG |
|---|---|---|---|---|---|---|---|
| **Model type** | / | / | **VPSDE** | **VPSDE** | **VPSDE** | **GVP** | **GVP** |
| **Function type** | / | / | $\epsilon(x_t, t)$ | $\epsilon(x_t, t)$ | $\epsilon(x_t, t)$ | $v(x_t, t)$ | $v(x_t, t)$ |
| **Pretrain scheme** | / | / | Eq. 2 | Eq. 2 | Eq. 2 | / | Eq. 3 |
| **Fintune scheme** | / | / | Eq. 30 | / | Eq. 35 | Eq. 40 | Eq. 8 |
| Cartpole swingup | $856 \pm 13$ | $798 \pm 31$ | $806 \pm 54$ | $851 \pm 9$ | $842 \pm 13$ | $830 \pm 36$ | $858 \pm 51$ |
| Cheetah run | $308 \pm 122$ | $304 \pm 32$ | $338 \pm 135$ | $451 \pm 231$ | $344 \pm 127$ | $359 \pm 188$ | $503 \pm 212$ |
| Humanoid run | $1.72 \pm 1.66$ | $303 \pm 6$ | $245 \pm 45$ | $179 \pm 91$ | $242 \pm 22$ | $226 \pm 72$ | $209 \pm 61$ |
| Manipulator insert ball | $154 \pm 55$ | $409 \pm 5$ | $340 \pm 451$ | $308 \pm 433$ | $352 \pm 458$ | $402 \pm 489$ | $686 \pm 341$ |
| Walker stand | $930 \pm 46$ | $689 \pm 14$ | $672 \pm 266$ | $850 \pm 161$ | $946 \pm 23$ | $593 \pm 287$ | $771 \pm 292$ |
| Finger turn hard | $714 \pm 80$ | $433 \pm 3$ | $698 \pm 352$ | $534 \pm 417$ | $328 \pm 417$ | $738 \pm 204$ | $657 \pm 371$ |
| Fish swim | $180 \pm 55$ | $504 \pm 13$ | $412 \pm 297$ | $474 \pm 248$ | $597 \pm 356$ | $634 \pm 192$ | $515 \pm 168$ |
| Manipulator insert peg | $50.4 \pm 9.2$ | $290 \pm 15$ | $279 \pm 229$ | $314 \pm 376$ | $327 \pm 383$ | $398 \pm 481$ | $540 \pm 343$ |
| Walker walk | $549 \pm 366$ | $651 \pm 8$ | $791 \pm 150$ | $887 \pm 51$ | $963 \pm 15$ | $869 \pm 241$ | $656 \pm 233$ |
| **Average** | $416 \pm 83$ | $487 \pm 14$ | $509 \pm 220$ | $538 \pm 224$ | $549 \pm 207$ | $561 \pm 243$ | $599 \pm 230$ |

Figure 1 illustrates the log-likelihood of D4RL datasets evaluated by GMPO/GMPG policies across different training iterations. A higher log-likelihood indicates that the generative model output is closer to the original data distribution. For hopper-medium-v2, the generative model trained with GMPO maintains a certain distance from the original data distribution throughout training. In contrast, the GMPG-trained model closely aligns with the original data distribution during pretraining but diverges more during finetuning compared to GMPO. This divergence allows GMPG to achieve better performance. In the case of halfcheetah-medium-expert-v2, both GMPO and GMPG benefit from high-quality data, as the optimal policy is already near the original distribution. Here, GMPO excels in filtering out high Q-value actions, resulting in a slightly better performance compared to GMPG.

More experiment details for GMPO and GMPG on D4RL AntMaze dataset can be found in Table 11 (Appendix D).

In general, GMPG with reverse KL loss outperforms GMPO and other generative policies with Forward KL loss in most medium and medium-replay locomotion tasks. This suggests that the policy gradient method more aggressively leverages Q guidance and is less constrained by the behavior policy, allowing optimization into regions with less data support — a beneficial exploration strategy in medium and medium-replay data, though it results in slightly poorer performance with expert data as shown in Table 3. Additionally, GMPO shows more stable training convergence with monotonic improvement, while GMPG exhibits fluctuating performance during training with small batch sizes, requiring larger batch sizes for stability.

## 6.3 ABLATION EXPERIMENTS

**Generative Model Type.** Table 5 presents a performance comparison of three generative models: GVP, VPSDE, and I-CFM (a more detailed comparison is available in Table 10). Overall, all three models demonstrate comparable performance. However, I-CFM exhibits slightly weaker performance in certain cases. This discrepancy may stem from its simpler flow path, as defined in Eq. 28, which could potentially limit its ability to capture the environment's complex dynamics effectively.

**Sampling Scheme for GMPG.** As shown in Table 6, our ablation study reveals that increasing the number of sampling time steps $T$ within the GMPG algorithm can lead to improved performance.

Further ablation experiments about temperature coefficient $\beta$ and solver schemes for sampling are detailed in Appendix D.3.

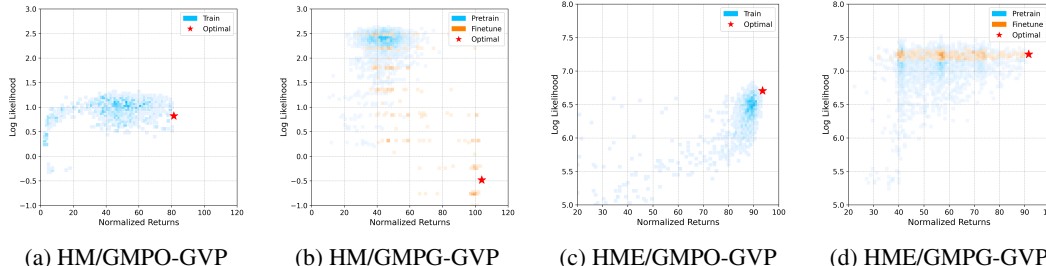

| (a) HM/GMPO-GVP | (b) HM/GMPG-GVP | (c) HME/GMPO-GVP | (d) HME/GMPG-GVP |

Figure 1: Log-Likelihood of D4RL datasets evaluated by GMPO/GMPG-GVP policies during training. HM stands for hopper-medium-v2, HME stands for halfcheetah-medium-expert-v2. Each point represents a model during training, with colors indicating different stages. The returns of the model are evaluated and averaged over five random seeds. Blue points denote the pretraining stage for GMPG and the training stage for GMPO, as GMPO does not require pretraining. Orange points indicate the finetuning stage for GMPG. The star marker shows the optimal model obtained during training. The density of the points reflects the number of models in that area.

Table 5: Performance comparison of model types on D4RL datasets. The average performance is calculated over 9 locomotion tasks.

| Algo. type | GMPO | | | GMPG | | |
|---|---|---|---|---|---|---|
| Model type | VPSDE | GVP | I-CFM | VPSDE | GVP | I-CFM |
| Average (Locomotion) | $80.2 \pm 4.2$ | $82.7 \pm 4.8$ | $76.2 \pm 8.0$ | $87.3 \pm 3.5$ | $84.2 \pm 3.2$ | $83.4 \pm 4.2$ |

Table 6: Performance comparison of different sampling time steps for GMPG. The time span of the ODE solver is $[0, 1]$, so a larger $T$ means more sampling steps and a smaller step size.

| Pretrain scheme / Finetune scheme | GMPG / VPSDE / $v(x_t, t)$ Eq. 3 / Eq. 8 | | |
|---|---|---|---|
| $T$ (Used for Training Only) | 32 | 100 | 1000 |
| halfcheetah-medium-v2 | $53.9 \pm 2.7$ | $55.8 \pm 2.8$ | $57.0 \pm 3.1$ |
| halfcheetah-medium-replay-v2 | $43.4 \pm 3.5$ | $49.1 \pm 3.3$ | $50.5 \pm 2.7$ |

## 7 CONCLUSION

In this paper, we provide a comprehensive study of generative policies and propose two unified and RL-native training schemes, GMPO and GMPG, that are effective and straightforward for both diffusion and flow models. GMPO benefits from a stable training process and does not require pretraining the generative model before optimal policy extraction, making it more efficient and easier to train. GMPG is a native policy gradient-based method for continuous-time generative models, and we provide a numerically stable implementation for the RL community. Our experiment results demonstrate that the proposed training schemes offer comparable or better performance than previous works in most cases.

To ensure consistent comparisons and analyses, we introduce a unified framework for reinforcement learning algorithms that leverages the expressive power of generative models. This standardized experimental framework decouples the generative model from the RL components, allowing for consistent evaluation of different generative models within the same RL context.

Overall, this work simplifies and unifies the training of generative models for policy modeling, providing practical guidelines for training and deploying generative policies in reinforcement learning. Additionally, we discuss existing limitations and valuable topics for future work in Appendix F.

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

## A  POLICY OPTIMIZATION

We provide a detailed derivation of the forward KL and reverse KL divergence training objectives in Appendix A.1, and discuss the theoretical connections and differences between these objectives in Appendix A.2.

### A.1  DERIVATION DETAILS

The full derivation of the forward KL divergence training objective in Eq. 4 is as follows:

$$
\begin{aligned}
\mathcal{L}(\theta) &= \mathbb{E}_{s \sim \mathcal{D}} \left[ D_{\mathrm{KL}} \left[ \pi^*(\cdot|s) \| \pi_\theta(\cdot|s) \right] \right] \\
&= \mathbb{E}_{s \sim \mathcal{D}} \left[ \int \pi^*(a|s)(\log \pi^*(a|s) - \log \pi_\theta(a|s)) \mathrm{d}a \right] \\
&= \mathbb{E}_{s \sim \mathcal{D}} \left[ \int \pi^*(a|s) \log \pi^*(a|s) \mathrm{d}a - \int \pi^*(a|s) \log \pi_\theta(a|s) \mathrm{d}a \right] \\
&= \mathbb{E}_{s \sim \mathcal{D}} \left[ -\mathcal{H}_{\pi^*(a|s)} - \int \frac{e^{\beta(Q(s,a)-V(s))}}{Z(s)} \mu(a|s) \log \pi_\theta(a|s) \mathrm{d}a \right] \\
&= \mathbb{E}_{s \sim \mathcal{D}, a \sim \mu(\cdot|s)} \left[ -\frac{e^{\beta(Q(s,a)-V(s))}}{Z(s)} \log \pi_\theta(a|s) \right] - \mathbb{E}_{s \sim \mathcal{D}} \left[ \mathcal{H}_{\pi^*(a|s)} \right] \\
&= \mathbb{E}_{s \sim \mathcal{D}, a \sim \mu(\cdot|s)} \left[ -\frac{e^{\beta(Q(s,a)-V(s))}}{Z(s)} \log \pi_\theta(a|s) \right] + C.
\end{aligned}
\tag{9}
$$

The full derivation of the reverse KL divergence training objective in Eq. 5 is as follows:

$$
\begin{aligned}
\mathcal{L}(\theta) &= \mathbb{E}_{s \sim \mathcal{D}} \left[ D_{\mathrm{KL}} \left[ \pi_\theta(\cdot|s) \| \pi^*(\cdot|s) \right] \right] \\
&= \mathbb{E}_{s \sim \mathcal{D}} \left[ \int \pi_\theta(a|s)(\log \pi_\theta(a|s) - \log \pi^*(a|s)) \mathrm{d}a \right] \\
&= \mathbb{E}_{s \sim \mathcal{D}} \left[ \int \pi_\theta(a|s)(\log \pi_\theta(a|s) - \log \left( \frac{e^{\beta(Q(s,a)-V(s))}}{Z(s)} \mu(a|s) \right)) \mathrm{d}a \right] \\
&= \mathbb{E}_{s \sim \mathcal{D}} \left[ \int \pi_\theta(a|s)(\log \pi_\theta(a|s) - \log \mu(a|s) - \log \left( \frac{e^{\beta(Q(s,a)-V(s))}}{Z(s)} \right)) \mathrm{d}a \right] \\
&= \mathbb{E}_{s \sim \mathcal{D}} \left[ D_{\mathrm{KL}} \left[ \pi_\theta(\cdot|s) \| \mu(\cdot|s) \right] + \int -\pi_\theta(a|s) \log \left( \frac{e^{\beta(Q(s,a)-V(s))}}{Z(s)} \right) \mathrm{d}a \right] \\
&= \mathbb{E}_{s \sim \mathcal{D}} \left[ D_{\mathrm{KL}} \left[ \pi_\theta(\cdot|s) \| \mu(\cdot|s) \right] + \int -\pi_\theta(a|s)(\beta Q(s,a) - \beta V(s) - \log Z(s)) \mathrm{d}a \right] \\
&= \mathbb{E}_{s \sim \mathcal{D}} \left[ D_{\mathrm{KL}} \left[ \pi_\theta(\cdot|s) \| \mu(\cdot|s) \right] \int \pi_\theta(a|s) \mathrm{d}a - \int \pi_\theta(a|s) \beta Q(s,a) \mathrm{d}a + (\beta V(s) + \log Z(s)) \int \pi_\theta(a|s) \mathrm{d}a \right] \\
&= \mathbb{E}_{s \sim \mathcal{D}} \left[ \int \pi_\theta(a|s)(-\beta Q(s,a) + D_{\mathrm{KL}} \left[ \pi_\theta(\cdot|s) \| \mu(\cdot|s) \right]) \mathrm{d}a + (\beta V(s) + \log Z(s)) \int \pi_\theta(a|s) \mathrm{d}a \right] \\
&= \mathbb{E}_{s \sim \mathcal{D}} \left[ \int \pi_\theta(a|s)(-\beta Q(s,a) + D_{\mathrm{KL}} \left[ \pi_\theta(\cdot|s) \| \mu(\cdot|s) \right]) \mathrm{d}a + \beta V(s) + \log Z(s) \right] \\
&= \mathbb{E}_{s \sim \mathcal{D}} \left[ \int \pi_\theta(a|s)(-\beta Q(s,a) + D_{\mathrm{KL}} \left[ \pi_\theta(\cdot|s) \| \mu(\cdot|s) \right]) \mathrm{d}a \right] + \mathbb{E}_{s \sim \mathcal{D}} \left[ \beta V(s) + \log Z(s) \right] \\
&= \mathbb{E}_{s \sim \mathcal{D}, a \sim \pi_\theta(\cdot|s)} \left[ -\beta Q(s,a) + D_{\mathrm{KL}} \left[ \pi_\theta(\cdot|s) \| \mu(\cdot|s) \right] \right] + C.
\end{aligned}
\tag{10}
$$

## A.2 COMPARISON BETWEEN FORWARD AND REVERSE KL DIVERGENCE TRAINING OBJECTIVES

To obtain a neural network approximation $\pi_\theta(a|s)$ of the optimal policy

$$\pi^*(a|s) = \frac{e^{\beta(Q(s,a)-V(s))}}{Z(s)}\mu(a|s), \tag{11}$$

where the policy is a state-conditioned probability distribution, we can use either the forward KL divergence or reverse KL divergence as the training objective. Minimizing the KL divergence between $\pi_\theta(a|s)$ and the optimal policy $\pi^(a|s)$ reduces the discrepancy between the two distributions:

$$\text{Minimize} \quad D_{\text{KL}}\left[\pi^*(\cdot|s)\|\pi_\theta(\cdot|s)\right] \quad \text{or} \quad D_{\text{KL}}\left[\pi_\theta(\cdot|s)\|\pi^*(\cdot|s)\right]. \tag{12}$$

However, as is well-known in the literature (Murphy, 2023), the forward and reverse KL divergence objectives have different properties and implications. Training with the forward KL divergence tends to be *mode-covering*, preferring to cover all modes of the target distribution, including those with low probability mass. In contrast, training with the reverse KL divergence tends to be *mode-seeking*, concentrating on modes with high probability mass while potentially ignoring others.

Although we employ highly expressive probabilistic models, such as diffusion and flow models, which may mitigate these tendencies, the two training objectives still have different implications and may lead to different performance in practice. We provide a detailed term-by-term illustration of these two methods and their variants in Appendix A.2.1 and A.2.2, respectively.

### A.2.1 FORWARD KL DIVERGENCE TRAINING OBJECTIVE

As shown in Eq. 13, the forward KL divergence training objective involves three components:

- The behavior policy $\mu(\cdot|s)$, from which actions are sampled.
- An exponential function of the advantage, $e^{\beta(Q(s,a)-V(s))}$, acting as an importance weight.
- The log-likelihood term $\log \pi_\theta(a|s)$ for maximum likelihood estimation (MLE).

$$\mathcal{L}(\theta) = \underbrace{\mathbb{E}_{s\sim\mathcal{D},a\sim\mu(\cdot|s)}}_{\text{Behavior Policy}}\left[-\underbrace{\frac{e^{\beta(Q(s,a)-V(s))}}{Z(s)}}_{\text{Advantage Weights}}\underbrace{\log \pi_\theta(a|s)}_{\text{MLE}}\right] + C. \tag{13}$$

In this objective, actions are sampled from the behavior policy $\mu(\cdot|s)$, which serves as a strong prior since it is static during optimization and pretrained to a well-established stage. Consequently, the range of actions evaluated is stable and rarely extends beyond the support of the behavior policy.

The exponential advantage term in Eq. 13 acts as an importance weight, emphasizing actions with higher advantage values. This term is crucial, as it focuses the learning on actions that improve policy performance. By weighting actions according to their advantages, suboptimal actions are filtered out in favor of optimal ones. If the advantage approaches zero for all actions, the objective simplifies to standard MLE.

In practice, some algorithms use explicit training objectives with advantage weights, such as QGPO (Lu et al., 2023) and GMPO, while others incorporate advantage weights during inference, such as SfBC (Chen et al., 2023) and IDQL (Hansen-Estruch et al., 2023), essentially making them sampling-based approaches when extracting optimal actions.

The MLE term $\log \pi_\theta(a|s)$ in Eq. 13 realizes the maximization of likelihood. However, as noted by Finlay et al. (2020), directly using the log-likelihood term for policy optimization with flow-based models can lead to ill-posed generative trajectories and poor performance. This is due to the infinite number of generation trajectories that can bridge two probability distributions, making the obtained trajectories sensitive.

Therefore, diffusion or flow models based on score matching or flow matching methods, which have static generation trajectories, are preferred over flow models trained by directly maximizing

log-likelihood. From a theoretical perspective, Song et al. (2021a) illustrated that conducting score matching with $\lambda(t) = g^2(t)$ is equivalent to training with an ELBO for maximizing likelihood estimation in diffusion models:

$$\mathcal{L}_{\text{MLE}}(\theta) = \mathbb{E}_{p(x)}\left[-\log p_\theta(x)\right] \leq \mathcal{L}_{\text{DSM}}(\theta) + C. \tag{14}$$

Recently, Lu et al. (2022) provided a more rigorous bound on how training diffusion models with score matching improves the log-likelihood, and Zheng et al. (2023a) presented similar proofs for flow matching methods. In general, any generative model training scheme that increases log-likelihood can serve as a substitute for $\log \pi_\theta$ in Eq. 13. Previous methods such as SfBC, IDQL, and QGPO use the score matching loss $\mathcal{L}_{\text{DSM}}(\theta)$, while in GMPO, we generalize this to include both the score matching loss $\mathcal{L}_{\text{DSM}}(\theta)$ and the flow matching loss $\mathcal{L}_{\text{CFM}}(\theta)$, collectively denoted as the matching loss $\mathcal{L}_{\text{Matching}}(\theta)$.

### A.2.2 REVERSE KL DIVERGENCE TRAINING OBJECTIVE

As shown in Eq. 15, the reverse KL divergence training objective comprises three components:

- The optimized policy $\pi_\theta(\cdot|s)$, from which actions are sampled.
- A value function $Q(s,a)$ that guides the policy optimization.
- A KL divergence term $D_{\text{KL}}(\pi_\theta(\cdot|s)|\mu(\cdot|s))$ acting as a proximal constraint.

$$\mathcal{L}(\theta) = \underbrace{\mathbb{E}_{s\sim\mathcal{D},a\sim\pi_\theta(\cdot|s)}}_{\text{Optimized Policy}}\left[-\underbrace{\beta Q(s,a)}_{\text{Guidance Function}} + \underbrace{D_{\text{KL}}(\pi_\theta(\cdot|s)\|\mu(\cdot|s))}_{\text{Proximal Constraint}}\right] + C. \tag{15}$$

In this objective, actions are sampled from the policy being optimized, $\pi_\theta(\cdot|s)$, which is initialized to the behavior policy $\mu(\cdot|s)$ but gradually diverges from it as optimization progresses guided by the value function. The value function $Q(s,a)$ provides guidance by emphasizing actions with higher values, encouraging the policy to focus on optimal actions, even if they are rare in the behavior policy. This contrasts with the forward KL objective, where actions are sampled from the behavior policy.

The proximal constraint term $D_{\text{KL}}(\pi_\theta(\cdot|s)|\mu(\cdot|s))$ enforces closeness to the behavior policy to prevent drastic updates that may degrade performance. While GMPG retains this KL divergence term, other algorithms replace it with alternative regularization methods, such as the score matching loss in Diffusion-QL (Wang et al., 2023) or a score regularization term in SRPO (Chen et al., 2024).

## B GENERATIVE MODELS

Generative models generate samples from a target distribution $p(x)$, or $p(x|c)$ when conditioned on context $c$. This work focuses on two types of continuous-time generative models: diffusion models and flow models.

### B.1 DIFFUSION MODELS

Diffusion models use a forward diffusion process to train the score function and a reverse diffusion process for sampling. Given a fixed data or target distribution, a diffusion model is determined by the diffusion process path. A common path, governed by a linear stochastic differential equation (SDE), is:

$$\mathrm{d}x = f(t)x_t\mathrm{d}t + g(t)\mathrm{d}w_t. \tag{16}$$

The transition distribution of a point $x \in \mathbb{R}^d$ from time 0 to $t$ is:

$$p(x_t|x_0) \sim \mathcal{N}(x_t|\alpha_t x_0, \sigma_t^2 I). \tag{17}$$

The drift coefficient $f(t)$ and diffusion coefficient $g(t)$ are related to the noise level $\sigma_t$ and scale level $\alpha_t$:

$$f(t) = \frac{\mathrm{d}\log\alpha_t}{\mathrm{d}t}, \tag{18}$$

$$g^2(t) = \frac{\mathrm{d}\sigma_t^2}{\mathrm{d}t} - 2\frac{\mathrm{d}\log\alpha_t}{\mathrm{d}t}\sigma_t^2. \tag{19}$$

The drift and diffusion coefficients for Linear Variance-Preserving SDE (VP-SDE) model (Song et al., 2021b) and Generalized VP-SDE (GVP) model (Albergo & Vanden-Eijnden, 2023) are defined as:

$$\text{VP-SDE:} \quad \alpha_t = \exp\left(-\frac{1}{2}\int_0^t \beta_s\,\mathrm{d}s\right), \qquad \sigma_t = \sqrt{1 - \exp\left(-\int_0^t \beta_s\,\mathrm{d}s\right)}.$$

$$\text{GVP:} \quad \alpha_t = \cos\left(\frac{1}{2}\pi t\right), \qquad\qquad \sigma_t = \sin\left(\frac{1}{2}\pi t\right). \tag{20}$$

Here, $\beta_t$ follows the same scaling as in Song et al. (2021b).

The reverse process of the diffusion model is derived from the Fokker-Planck equation and can be expressed as an ODE:

$$\frac{\mathrm{d}x_t}{\mathrm{d}t} = v(x_t) = f(t)x_t - \frac{1}{2}g^2(t)\nabla_{x_t}\log p(x_t), \tag{21}$$

where $v(x_t)$ is the velocity function and $\nabla_{x_t}\log p(x_t)$ is the score function, typically modeled by a neural network with parameters $\theta$, denoted as $v_\theta(x_t)$ and $s_\theta(x_t)$, respectively.

Training the diffusion model involves a matching objective $\mathcal{L}_{\text{Matching}}(\theta)$, which can utilize either the Score Matching method by Hyvärinen (2005) or the Flow Matching method by Lipman et al. (2023). The Score Matching objective is defined as a weighted Mean Squared Error (MSE) loss between the score function model and the gradient of the log density of the target distribution:

$$\mathcal{L}_{\text{SM}} = \frac{1}{2}\int_0^1 \mathbb{E}_{p(x_t)}\left[\lambda(t)\|s_\theta(x_t) - \nabla_{x_t}\log p(x_t)\|^2\right]\mathrm{d}t. \tag{22}$$

In practice, the Denoising Score Matching loss $\mathcal{L}_{\text{DSM}}$ is used for every diffusion path conditioned on $x_0$ because it shares the same gradient, $\nabla\mathcal{L}_{\text{DSM}} = \nabla\mathcal{L}_{\text{SM}}$, as shown by Vincent (2011):

$$\mathcal{L}_{\text{DSM}} = \frac{1}{2}\int_0^1 \mathbb{E}_{p(x_t,x_0)}\left[\lambda(t)\|s_\theta(x_t) - \nabla_{x_t}\log p(x_t|x_0)\|^2\right]\mathrm{d}t. \tag{23}$$

We can use both Vanilla Score Matching proposed by Ho et al. (2020) and Maximum Likelihood Score Matching by Song et al. (2021a). These methods differ in the weighting of the score matching loss:

$$\text{Vanilla Score Matching:} \quad \lambda_{\text{SM}}(t) = \sigma_t^2.$$

$$\text{Maximum Likelihood Score Matching:} \quad \lambda_{\text{MLSM}}(t) = g^2(t). \tag{24}$$

The Flow Matching method uses a weighted Mean Squared Error (MSE) loss between the velocity function of the reverse diffusion process and a target velocity:

$$\mathcal{L}_{\text{FM}} = \frac{1}{2}\int_0^1 \mathbb{E}_{p(x_t)}\left[\|v_\theta(x_t) - v(x_t)\|^2\right]\mathrm{d}t. \tag{25}$$

In practice, a Conditional Flow Matching loss $\mathcal{L}_{\text{CFM}}$ for every flow path conditioned on $x_0$ and $x_1$ is used, as it shares the same gradient, $\nabla\mathcal{L}_{\text{CFM}} = \nabla\mathcal{L}_{\text{FM}}$, as shown by Lipman et al. (2023) and Tong et al. (2024):

$$\mathcal{L}_{\text{CFM}} = \frac{1}{2}\int_0^1 \mathbb{E}_{p(x_t,x_0,x_1)}\left[\|v_\theta(x_t) - v(x_t|x_0,x_1)\|^2\right]\mathrm{d}t. \tag{26}$$

We use both the Score Matching loss $\mathcal{L}_{\text{DSM}}$ and the Flow Matching loss $\mathcal{L}_{\text{CFM}}$ to train the diffusion models.

## B.2 FLOW MODELS

Continuous normalizing flows (CNFs), introduced by Chen et al. (2018) and Grathwohl et al. (2019), are the first continuous-time generative models capable of modeling complex target distributions. However, their simulation-based maximum likelihood training process is unstable, often resulting in poor performance and distorted generative paths, as illustrated by Finlay et al. (2020).

To address these issues, recent works (Lipman et al. (2023); Liu et al. (2023); Pooladian et al. (2023); Albergo & Vanden-Eijnden (2023); Tong et al. (2024)) propose CNFs with simulation-free objectives, similar to diffusion models that use designed and fixed diffusion paths as regression objectives. These approaches are more stable during training and retain the advantage that the source distribution can be arbitrary, unlike diffusion models which require a Gaussian source distribution.

For the flow models I-CFM (Tong et al., 2024) investigated in this paper, the flow path is defined as:
$$p(x_t|x_0, x_1) = \mathcal{N}(x_t|tx_1 + (1 - t)x_0, \sigma^2 I). \tag{27}$$
The velocity function is given by:
$$v(x_t|x_0, x_1) = x_1 - x_0. \tag{28}$$
The flow model $v_\theta(x_t)$ is trained using the Conditional Flow Matching loss $\mathcal{L}_{\text{CFM}}$.

## C GENERATIVE POLICIES

As shown in Table 2, generative policy training schemes can be categorized based on their use of either the forward KL divergence (Eq. 4) or the reverse KL divergence (Eq. 5). SfBC and QGPO use forward KL divergence, while Diffusion-QL and SRPO employ reverse KL divergence. IDQL also uses forward KL divergence but differs slightly due to its unique importance weighting for sampling.

### C.1 SFBC: SELECTING FROM BEHAVIOR CANDIDATES

SfBC (Chen et al., 2023) trains a diffusion model as the behavior policy using the score matching loss (Eq. 2) and trains the Q-function model with an In-Support Q-Learning method (Eq. 29):

$$\mathcal{L}_{\text{in-support QL}}(\xi) = \mathbb{E}_{(s,a,s')\sim\mathcal{D}, a_i'\sim\mu} \left[ \left( Q_\xi(s,a) - r(s,a) - \gamma \left[ \frac{\sum_{i=1}^N e^{Q_\xi(s',a_i')} Q_\xi(s',a_i')}{\sum_{i=1}^N e^{Q_\xi(s',a_i')}} \right] \right)^2 \right].$$
$$\tag{29}$$

Actions are sampled by selecting the one with the highest Q-value among $N$ candidates generated by the behavior policy.

### C.2 QGPO: Q-GUIDED POLICY OPTIMIZATION

QGPO (Lu et al., 2023) enhances SfBC by distilling the Q-function into an intermediate energy guidance model, $\mathcal{E}_\phi(s, a, t)$, using the Contrastive Energy Prediction (CEP) method:

$$\mathcal{L}_{\text{CEP}}(\phi) = -\mathbb{E}_{t,(s,a)\sim\mathcal{D}, a_i\sim\mu} \left[ \sum_{i=1}^N \frac{e^{\beta Q(s,a_i)}}{\sum_{i=1}^N e^{\beta Q(s,a_i)}} \log \frac{e^{\mathcal{E}_\phi(s,a_i,t)}}{\sum_{i=1}^N e^{\mathcal{E}_\phi(s,a_i,t)}} \right]. \tag{30}$$

This approach allows the optimal policy to be sampled using a combination of the score functions of the behavior policy and the energy guidance model:
$$\nabla_{a_t} \pi(a_t|s_t) = \nabla_{a_t} \log \mu(a_t|s_t) + \nabla_{a_t} \mathcal{E}_\phi(s_t, a_t, t). \tag{31}$$
However, this method is not suitable for flow models, as their score functions cannot be easily obtained.

### C.3 IDQL: IMPLICIT DIFFUSION Q-LEARNING

IDQL (Hansen-Estruch et al., 2023) employs Implicit Q-Learning as described in Eq. 32:
$$\mathcal{L}_{\text{IQL-V}}(\psi) = \mathbb{E}_{(s,a)\sim\mathcal{D}} \left[ \mathcal{L}_2^\tau (Q_\xi(s,a) - V_\psi(s)) \right]$$
$$\mathcal{L}_2^\tau(u) = |\tau - \mathbb{1}(u \le 0)|u^2 \tag{32}$$
$$\mathcal{L}_{\text{IQL-Q}}(\xi) = \mathbb{E}_{(s,a)\sim\mathcal{D}} \left[ (Q_\xi(s,a) - r(s,a) - V_\psi(s'))^2 \right].$$

IDQL implicitly constructs an optimal policy model through importance sampling based on the Q-values of action candidates. A batch of backup actions is sampled from a behavior policy, and the optimal action is selected through inference using the Q-function model.

Hansen-Estruch et al. (2023) demonstrated that if an implicit actor satisfies the following equation:

$$V^*(s) = \mathbb{E}_{a \sim \pi_{\text{implicit}}(a|s)}[Q^*(s, a)], \tag{33}$$

for an optimal Q-function and value function in IQL, then the implicit actor is given by:

$$\pi_{\text{implicit}}(a|s) \propto \frac{\left| \frac{\partial}{\partial V(s)}(Q(s, a) - V^*(s)) \right|}{|Q(s, a) - V^*(s)|} \mu(a|s). \tag{34}$$

where $\frac{\left| \frac{\partial}{\partial V(s)}(Q(s,a) - V^*(s)) \right|}{|Q(s,a) - V^*(s)|}$ is the importance weight for action $a$ in state $s$.

### C.4 SRPO: Score-based Regularized Policy Optimization

SRPO (Chen et al., 2024) pretrains a behavior policy using a diffusion model with a score matching loss, as described in Eq. 2. The Q-function model is then independently trained using the IQL method outlined in Eq. 32.

The Dirac or Gaussian policy is trained by distilling guidance from the Q-function model, using the score function of the behavior policy as a regularizer to prevent significant divergence:

$$\mathcal{L}_{\text{SRPO}}(\psi) = -\beta Q(s, a) + w(t)(\epsilon_\psi(a_t|s) - \epsilon). \tag{35}$$

### C.5 Diffusion-QL

Diffusion-QL (Wang et al., 2023) trains the Q-function model using the conventional Bellman operator with the double Q-learning trick:

$$\mathcal{L}_{\text{Diffusion-QL}}(\xi) = \mathbb{E}_{(s,a,s') \sim \mathcal{D}, a' \sim \pi} \left[ \left( Q_\xi(s, a) - r(s, a) - \gamma \min_{i=1,2} Q_{\xi_i}(s', a') \right)^2 \right]. \tag{36}$$

Next, Q-value function guidance is incorporated into the policy using the following training objective:

$$\mathcal{L}_{\text{Diffusion-QL}}(\theta) = \mathbb{E}_{s \sim \mathcal{D}, a \sim \pi_\theta(\cdot|s)} \left[ -\beta Q(s, a) + \mathcal{L}_{\text{DSM}}(\theta) \right], \tag{37}$$

where $\mathcal{L}_{\text{DSM}}(\theta)$ is the score matching loss for the diffusion model, as defined in Eq. 2.

During inference, the authors adopt a resampling strategy similar to Implicit Diffusion Q-Learning, where the action with the highest Q-value is selected using a softmax function among $50$ candidates. A small performance drop is observed in Diffusion-QL when the resampling strategy is disabled, indicating that the policy model relies on the Q-function model to achieve optimal performance.

### C.6 GMPO: Generative Model Policy Optimization

The derivation of Eq. 6 is:

$$\begin{aligned}
\mathcal{L}_{\text{GMPO}}(\theta) &= \mathbb{E}_{s \sim \mathcal{D}, a \sim \pi^*(\cdot|s)} \left[ \mathcal{L}_{\text{Matching}}(\theta) \right] \\
&= \mathbb{E}_{s \sim \mathcal{D}} \left[ \int \pi^*(a|s) \mathcal{L}_{\text{Matching}}(\theta) \mathrm{d}a \right] \\
&= \mathbb{E}_{s \sim \mathcal{D}} \left[ \int \frac{e^{\beta(Q(s,a) - V(s))}}{Z(s)} \mu(a|s) \mathcal{L}_{\text{Matching}}(\theta) \mathrm{d}a \right] \\
&= \mathbb{E}_{s \sim \mathcal{D}} \left[ \int \mu(a|s) \frac{e^{\beta(Q(s,a) - V(s))}}{Z(s)} \mathcal{L}_{\text{Matching}}(\theta) \mathrm{d}a \right] \\
&= \mathbb{E}_{s \sim \mathcal{D}, a \sim \mu(\cdot|s)} \left[ \frac{e^{\beta(Q(s,a) - V(s))}}{Z(s)} \mathcal{L}_{\text{Matching}}(\theta) \right].
\end{aligned} \tag{38}$$

We present the GMPO algorithm in Algorithm 1.

---

**Algorithm 1** Generative Model Policy Optimization (GMPO)

---

    Initialize $\pi_\theta, Q_\xi, V_\phi$.
    *// Critic training (Implicit Q-Learning)*
    **for** epoch = 1 to $N$ **do**
        $\phi \leftarrow \phi - \lambda_Q \nabla_\phi L_V$            (Eq. 32)
        $\xi \leftarrow \xi - \lambda_V \nabla_\xi L_Q$            (Eq. 32)
    *// Policy training (Advantage-Weighted Regression)*
    **for** epoch = 1 to $N$ **do**
        $\theta \leftarrow \theta - \lambda_\pi \nabla_\theta L_\pi$            (Eq. 7)

---

For the score matching objective, the GMPO loss is defined as:

$$\mathcal{L}_{\text{GMPO-SM}}(\theta) = -\mathbb{E}_{s \sim \mathcal{D}, a \sim \mu(\cdot|s)} \left[ \frac{e^{\beta(Q(s,a)-V(s))}}{Z(s)} \mathbb{E}_{p(a_t|a)} \left[ \tfrac{1}{2}\lambda(t)\|s_\theta(a_t|s) - \nabla_{a_t} \log p(a_t|a,s)\|^2 \right] \right]. \tag{39}$$

For the flow matching objective, the GMPO loss is defined as:

$$\mathcal{L}_{\text{GMPO-FM}}(\theta) = -\mathbb{E}_{s \sim \mathcal{D}, a \sim \mu(\cdot|s)} \left[ \frac{e^{\beta(Q(s,a)-V(s))}}{Z(s)} \mathbb{E}_{p(a_t|a)} \left[ \frac{1}{2}\|v_\theta(a_t|s) - v(a_t|a,s)\|^2 \right] \right]. \tag{40}$$

To address potential numerical issues with the importance sampling weight in exponential form, we can clamp the weight if it becomes too large. This adjustment reduces the emphasis on high Q-value actions but does not significantly impact performance.

Alternatively, if clamping the weight is not desired, using a softmax function to approximate the importance weight can also circumvent numerical issues. This approach is similar to QGPO and is formulated as:

$$\mathcal{L}_{\text{GMPO-Softmax}}(\theta) = -\mathbb{E}_{s \sim \mathcal{D}, a_{1:K} \sim \mu(\cdot|s)} \left[ \frac{e^{\beta Q(s,a_i)}}{\sum_{j=1}^{K} e^{\beta Q(s,a_j)}} \mathcal{L}_{\text{Matching}}(\theta) \right]. \tag{41}$$

However, this requires sampling $K$ actions from the behavior policy, increasing computational cost during training.

We present the GMPO algorithm with the behavior policy in Algorithm 2.

---

**Algorithm 2** Generative Model Policy Optimization (GMPO) with Behavior Policy

---

    Initialize $\mu_{\theta_1}, \pi_{\theta_2}, Q_\xi, V_\phi$.
    *// Behavior Policy Pretraining (Score Matching or Flow Matching)*
    **for** epoch = 1 to $N$ **do**
        $\theta_1 \leftarrow \theta_1 - \lambda_\mu \nabla_{\theta_1} L_\mu$       (Eq. 2 or Eq. 3)
    *// Critic Training (Implicit Q-Learning)*
    **for** epoch = 1 to $N$ **do**
        $\phi \leftarrow \phi - \lambda_Q \nabla_\phi L_V$            (Eq. 32)
        $\xi \leftarrow \xi - \lambda_V \nabla_\xi L_Q$            (Eq. 32)
    *// Policy Training (Advantage-Weighted Regression)*
    **for** epoch = 1 to $N$ **do**
        Sample $a_i \sim \mu_{\theta_1}$
        $\theta_2 \leftarrow \theta_2 - \lambda_\pi \nabla_{\theta_2} L_\pi$            (Eq. 41)

---

## C.7   GMPG: Generative Model Policy Gradient

We present the GMPG algorithm in Algorithm 3.

---

**Algorithm 3** Generative Model Policy Gradient (GMPG)

---

Initialize $\mu_{\theta_1}, \pi_{\theta_2}, Q_\xi, V_\phi$.
*// Behavior Policy Pretraining (Flow Matching)*
**for** epoch = 1 to $N$ **do**
$\quad \theta_1 \leftarrow \theta_1 - \lambda_\mu \nabla_{\theta_1} L_\mu$ $\hfill$ (Eq. 3)
*// Critic Training (Implicit Q-Learning)*
**for** epoch = 1 to $N$ **do**
$\quad \phi \leftarrow \phi - \lambda_Q \nabla_\phi L_V$ $\hfill$ (Eq. 32)
$\quad \xi \leftarrow \xi - \lambda_V \nabla_\xi L_Q$ $\hfill$ (Eq. 32)
*// Policy Training (Policy Gradient)*
$\theta_2 \leftarrow \theta_1$
**for** epoch = 1 to $N$ **do**
$\quad \theta_2 \leftarrow \theta_2 - \lambda_\pi \nabla_{\theta_2} L_\pi$ $\hfill$ (Eq. 8 or Eq. 42)

---

Since the vanilla GMPG loss relies on sampling from a dynamic generative model, policy training can become unstable with a small batch size, particularly when the optimized policy encounters state-action spaces with scarce data.

To stabilize the training process, we can use an importance sampling weight by sampling from a static generative model, which in practice is the behavior policy $\mu(\cdot|s)$:

$$\nabla_\theta \mathcal{L}_{\text{GMPG-Static}}(\theta) =$$

$$\mathbb{E}_{s\sim\mathcal{D},a\sim\mu(\cdot|s)} \left[ \frac{e^{\beta(Q(s,a)-V(s))}}{Z(s)} (-\beta Q(s,a) + \log \pi_\theta(a|s) - \log \mu(a|s)) \nabla_\theta \log \pi_\theta(a|s) \right]$$

$$\mathbb{E}_{s\sim\mathcal{D},a_{1:K}\sim\mu(\cdot|s)} \left[ \frac{e^{\beta Q(s,a_i)}}{\sum_{j=1}^K e^{\beta Q(s,a_j)}} (-\beta Q(s,a_i) + \log \pi_\theta(a_i|s) - \log \mu(a_i|s)) \nabla_\theta \log \pi_\theta(a_i|s) \right].$$
$$\tag{42}$$

It is important to note that GMPG works well only when the generative model's neural network output is a velocity function $v_\theta$. Training encounters numerical issues when using a noise function $\epsilon_\theta$ as the neural network output. If using a neural network as the noise function, converting it to a velocity function by Eq 21:

$$v_\theta(x_t) = f(t)x_t - \frac{1}{2}g^2(t)\nabla_{x_t} \log p_\theta(x_t) = f(t)x_t - \frac{g^2(t)}{2\sigma(t)}\epsilon_\theta(x_t). \tag{43}$$

As $g^2(t)$ approaches 0 when sampling from $t = 1$ to $t = 0$, $\sigma(t)$ approching 0 much faster than $g^2(t)$. Therefore, since the noise function $\epsilon_\theta$ is a neural network output and is in the range $[-D, D]$, the velocity can become unstable. Although this is not a major issue during sampling $dx = vdt$, since $dt$ is not 0 for ODE solvers, it can cause numerical instability during training if using neural ODE implementation for backward gradient computation. Therefore, we use only the velocity model for GMPG experiments across different generative models.

### C.7.1 CALCULATION OF PROBABILITY DENSITY

The calculation of $\log p(x)$ is crucial for training the generative model via the GMPG algorithm, following the approach of Chen et al. (2018); Grathwohl et al. (2019). Since sampling through the generative model involves solving an ODE defined by Eq. 21, the probability density of the sampled variable $x_t$ can be computed using the instantaneous change of variables theorem:

$$\begin{bmatrix} x_0 \\ \log p(x_0) - \log p(x_1) \end{bmatrix} = \int_{t_1}^{t_0} \begin{bmatrix} v_\theta(x(t), t) \\ -\text{Tr}\left(\frac{\partial v_\theta}{\partial x(t)}\right) \end{bmatrix} dt. \tag{44}$$

Here, $x_1 \sim \mathcal{N}(0, I)$ and $\log p(x_1)$ is tractable. Thus, $\log p(x_0)$ can be obtained by integrating from $t_1 = 1$ to $t_0 = 0$.

To reduce the computational cost of calculating the trace of the Jacobian matrix, we use Hutchinson's trace estimator (Hutchinson, 1990), which approximates the trace as:

$$\text{Tr}\left(\frac{\partial v_\theta}{\partial x(t)}\right) \approx \mathbb{E}_{\epsilon \sim \mathcal{N}(0,I)}\left[\epsilon^\top \frac{\partial v_\theta}{\partial x(t)}\epsilon\right], \tag{45}$$

where $\epsilon \sim \mathcal{N}(0, I)$ is a random vector sampled from a standard Gaussian distribution.

### C.8  2D TOY EXAMPLE

To visually illustrate the fundamental differences between the proposed generative policies, we use a simple 2D toy example with the Swiss Roll dataset. We designed a value function for this dataset, where the value changes from $-3.5$ to $1.5$ as the spiral extends outward (see Figure 2).

We evaluate the generation trajectories of models trained with GMPO and GMPG on this example, with results shown in Figure 3. This demonstration highlights that GMPO and GMPG operate differently: GMPO filters data points and learns the path to these points, whereas GMPG attempts to keep the generation path within the manifold of the original data distribution. Consequently, the Swiss Roll shape is largely preserved in GMPG trajectories but not in GMPO trajectories.

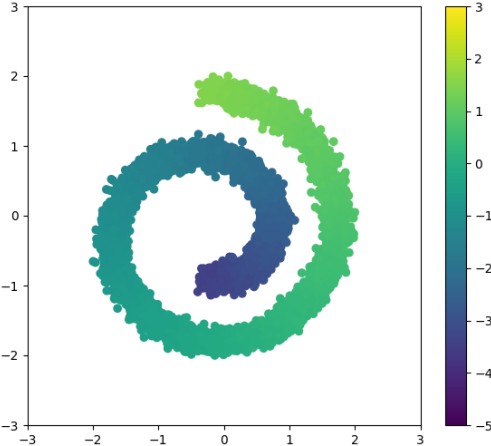

Figure 2: 2D toy Swiss Roll dataset with assigned value function. Values range from $-3.5$ to $1.5$ as the spiral extends outward. Colors represent data point values. A small noise $\epsilon = 0.6$ is added for better visualization.

## D  EXPERIMENTS

### D.1  TRAINING DETAILS

Training is conducted on an NVIDIA A100 GPU with 80GB of memory. The duration for each experiment ranges from 24 to 72 hours, depending on the model complexity, dataset size, and chosen training steps. Table 7 lists the hyperparameters used for training generative models and reinforcement learning models.

We tune the temperature coefficient $\beta$ for different tasks, as it affects the strength of the Q-value guidance. Choosing the appropriate value is sometimes essential for optimal policy model performance, as shown in Table 8.

Table 7: Hyper-parameters for training generative models and reinforcement learning models

| Training | |
|---|---|
| Optimizer | Adam |
| $\tau$ in IQL | 0.7 |
| $\tau$ in IQL for AntMaze | 0.9 |
| Discount factor $\gamma$ | 0.99 |
| Learning rate for pretraining behavior model | $10^{-4}$ |
| Learning rate for critic training | $10^{-4}$ |
| Learning rate for policy extraction | $10^{-4}$ |
| Learning rate for policy extraction (special cases) | $10^{-5} \sim 10^{-6}$ |
| Batchsize for pretraining behavior model | 4096 |
| Batchsize for critic training | 4096 |
| Batchsize for policy extraction in GMPO | 4096 |
| Batchsize for policy extraction in GMPG | 40960 |
| Sampling steps $T$ | 1000 |
| Evaluation | |
| Solver for ODE | Euler-Maruyama |
| Sampling steps $T$ | 32 |

Table 8: Temperature coefficient $\beta$ value tuning over different tasks

| Task | GMPO | GMPG |
|---|---|---|
| D4RL Locomotion | | |
| halfcheetah-medium-expert-v2 | 1.0 | 4.0 |
| hopper-medium-expert-v2 | 1.0 | 4.0 |
| walker2d-medium-expert-v2 | 1.0 | 4.0 |
| halfcheetah-medium-v2 | 1.0 | 1.0 |
| hopper-medium-v2 | 16.0 | 20.0 |
| walker2d-medium-v2 | 8.0 | 1.0 |
| halfcheetah-medium-replay-v2 | 4.0 | 4.0 |
| hopper-medium-replay-v2 | 6.0 | 8.0 |
| walker2d-medium-replay-v2 | 8.0 | 4.0 |
| D4RL AntMaze | | |
| antmaze-umaze-v0 | 8.0 | 1.0 |
| antmaze-umaze-diverse-v0 | 16.0 | 1.0 |
| antmaze-medium-play-v0 | 12.0 | 0.25 |
| antmaze-medium-diverse-v0 | 12.0 | 0.25 |
| antmaze-large-play-v0 | 16.0 | 0.5 |
| antmaze-large-diverse-v0 | 4.0 | 1.0 |
| RL Unplugged DeepMind Control Suite | | |
| Cartpole swingup | 1.0 | 4.0 |
| Cheetah run | 1.0 | 4.0 |
| Humanoid run | 1.0 | 4.0 |
| Manipulator insert ball | 1.0 | 4.0 |
| Walker stand | 1.0 | 4.0 |
| Finger turn hard | 1.0 | 4.0 |
| Fish swim | 1.0 | 4.0 |
| Manipulator insert peg | 1.0 | 4.0 |
| Walker walk | 1.0 | 4.0 |

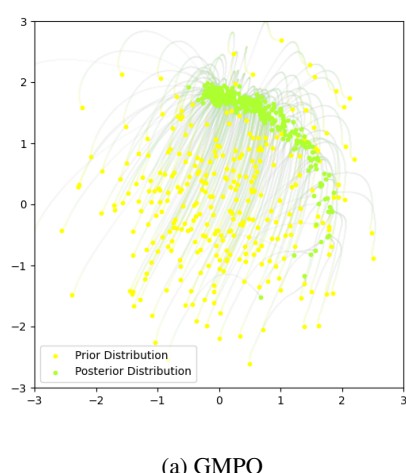

(a) GMPO

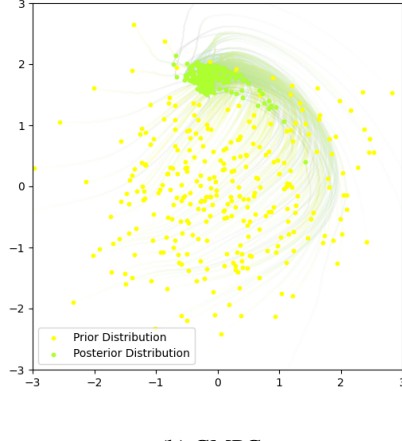

(b) GMPG

Figure 3: Generation trajectories of models trained by GMPO and GMPG on the 2D toy Swiss Roll dataset. Colors indicate time stamps of data points during generation.

Table 9: Performance comparison of on D4RL datasets across QGPO, SRPO and IDQL algorithms between GenerativeRL and original papers.

| | Original Papers | | | GenerativeRL | | |
|---|---|---|---|---|---|---|
| Algo. type | QGPO | IDQL | SRPO | QGPO | IDQL | SRPO |
| Model type | VPSDE | DDPM | VPSDE | VPSDE | VPSDE | VPSDE |
| Function type | $\epsilon(x_t, t)$ | $\epsilon(x_t, t)$ | $\epsilon(x_t, t)$ | $\epsilon(x_t, t)$ | $\epsilon(x_t, t)$ | $\epsilon(x_t, t)$ |
| Pretrain scheme | Eq. 2 | Eq. 2 | Eq. 2 | Eq. 2 | Eq. 2 | Eq. 2 |
| Finetune scheme | Eq. 30 | / | Eq. 35 | Eq. 30 | / | Eq. 35 |
| halfcheetah-medium-expert-v2 | 93.5 | 95.9 | 92.2 | $92.0 \pm 1.5$ | $91.7 \pm 2.4$ | $86.7 \pm 3.7$ |
| hopper-medium-expert-v2 | 108.0 | 108.6 | 100.1 | $107.0 \pm 0.9$ | $96.8 \pm 10.4$ | $100.8 \pm 9.3$ |
| walker2d-medium-expert-v2 | 110.7 | 112.7 | 114.0 | $107.3 \pm 1.3$ | $107.0 \pm 0.5$ | $118.7 \pm 1.4$ |
| halfcheetah-medium-v2 | 54.1 | 51.0 | 60.4 | $44.0 \pm 0.7$ | $43.7 \pm 2.8$ | $51.4 \pm 2.9$ |
| hopper-medium-v2 | 98.0 | 65.4 | 95.5 | $80.1 \pm 7.0$ | $72.1 \pm 17.6$ | $97.2 \pm 3.3$ |
| walker2d-medium-v2 | 86.0 | 82.5 | 84.4 | $82.8 \pm 2.7$ | $82.0 \pm 2.4$ | $85.6 \pm 2.1$ |
| halfcheetah-medium-replay-v2 | 47.6 | 45.9 | 51.4 | $42.5 \pm 1.7$ | $41.6 \pm 8.4$ | $47.2 \pm 4.5$ |
| hopper-medium-replay-v2 | 96.9 | 92.1 | 101.2 | $99.3 \pm 1.8$ | $89.1 \pm 3.1$ | $78.2 \pm 12.1$ |
| walker2d-medium-replay-v2 | 84.4 | 85.1 | 84.6 | $81.1 \pm 4.2$ | $80.4 \pm 9.2$ | $79.6 \pm 7.6$ |
| **Average (Locomotion)** | 86.6 | 82.1 | 87.1 | $81.8 \pm 2.4$ | $78.3 \pm 6.3$ | $82.8 \pm 5.1$ |

## D.2 MORE EXPERIMENTS

We provide comparative performance scores for the QGPO, SRPO, and IDQL algorithms on D4RL datasets, using both our implementation and the scores reported in the original papers, as shown in Table 9.

We conduct experiments on the D4RL to evaluate the performance of generative policy using different generative models as shown in Table 10.

We conducted additional experiments on the D4RL AntMaze datasets to evaluate the performance of GMPO and GMPG, as shown in Table 11. In this 2D maze environment, the agent navigates to a goal location, with a larger action and state space compared to previous D4RL locomotion experiments. Performance is evaluated based on the average return over 100 episodes. Our results indicate that the proposed generative policies, GMPO and GMPG, achieve competitive performance compared to the baselines.

Table 10: Performance comparison of on D4RL datasets over different generative models over GMPO and GMPG.

| Algo. type | GMPO | | | GMPG | | |
|---|---|---|---|---|---|---|
| Model type | VPSDE | GVP | I-CFM | VPSDE | GVP | I-CFM |
| Function type | $\epsilon(x_t,t)$ | $v(x_t,t)$ | $v(x_t,t)$ | $v(x_t,t)$ | $v(x_t,t)$ | $v(x_t,t)$ |
| Pretrain scheme | / | / | / | Eq. 3 | Eq. 3 | Eq. 3 |
| Fintune scheme | Eq. 39 | Eq. 40 | Eq. 40 | Eq. 8 | Eq. 8 | Eq. 8 |
| halfcheetah-medium-expert-v2 | $91.8 \pm 3.3$ | $91.9 \pm 3.2$ | $83.3 \pm 3.7$ | $89.0 \pm 6.4$ | $84.2 \pm 8.0$ | $86.9 \pm 4.5$ |
| hopper-medium-expert-v2 | $111.1 \pm 1.34$ | $112.0 \pm 1.8$ | $87.4 \pm 25.7$ | $107.8 \pm 1.9$ | $101.6 \pm 2.9$ | $101.7 \pm 1.4$ |
| walker2d-medium-expert-v2 | $107.7 \pm 0.4$ | $108.1 \pm 0.7$ | $110.3 \pm 0.7$ | $112.8 \pm 1.2$ | $110.0 \pm 1.2$ | $110.7 \pm 0.3$ |
| halfcheetah-medium-v2 | $49.8 \pm 2.6$ | $49.9 \pm 2.7$ | $48.0 \pm 2.9$ | $57.0 \pm 3.1$ | $46.0 \pm 2.7$ | $51.4 \pm 2.9$ |
| hopper-medium-v2 | $71.9 \pm 22.1$ | $74.6 \pm 21.2$ | $69.5 \pm 20.4$ | $101.1 \pm 2.6$ | $100.1 \pm 1.6$ | $92.8 \pm 18.1$ |
| walker2d-medium-v2 | $79.0 \pm 13.2$ | $81.1 \pm 4.3$ | $79.2 \pm 7.6$ | $91.9 \pm 0.9$ | $92.0 \pm 1.1$ | $82.6 \pm 2.3$ |
| halfcheetah-medium-replay-v2 | $36.6 \pm 2.4$ | $42.3 \pm 3.6$ | $41.7 \pm 3.2$ | $50.5 \pm 2.7$ | $39.1 \pm 5.4$ | $41.0 \pm 3.5$ |
| hopper-medium-replay-v2 | $89.2 \pm 7.4$ | $97.8 \pm 3.8$ | $86.0 \pm 2.6$ | $86.3 \pm 10.5$ | $103.4 \pm 2.1$ | $104.2 \pm 2.0$ |
| walker2d-medium-replay-v2 | $84.5 \pm 4.6$ | $86.4 \pm 1.7$ | $80.9 \pm 5.3$ | $90.1 \pm 2.2$ | $81.7 \pm 3.2$ | $79.4 \pm 3.2$ |
| **Average (Locomotion)** | $80.2 \pm 4.2$ | $82.7 \pm 4.8$ | $76.2 \pm 8.0$ | $87.3 \pm 3.5$ | $84.2 \pm 3.2$ | $83.4 \pm 4.2$ |

Table 11: Performance evaluation on D4RL AntMaze of different generative policies.

| Environment | SfBC | Diffusion-QL | QGPO | IDQL | SRPO | GMPO | GMPG |
|---|---|---|---|---|---|---|---|
| Model type | VPSDE | DDPM | VPSDE | DDPM | VPSDE | GVP | VPSDE |
| Function type | $\epsilon(x_t,t)$ | $\epsilon(x_t,t)$ | $\epsilon(x_t,t)$ | $\epsilon(x_t,t)$ | $\epsilon(x_t,t)$ | $v(x_t,t)$ | $v(x_t,t)$ |
| Pretrain scheme | Eq. 2 | Eq. 2 | Eq. 2 | Eq. 2 | Eq. 2 | / | Eq. 3 |
| Fintune scheme | / | Eq. 37 | Eq. 30 | / | Eq. 35 | Eq. 40 | Eq. 8 |
| antmaze-umaze-v0 | 92.0 | 93.4 | 96.4 | 94.0 | 97.1 | $94.2 \pm 0.9$ | $92.5 \pm 1.6$ |
| antmaze-umaze-diverse-v0 | 85.3 | 66.2 | 74.4 | 80.2 | 82.1 | $76.8 \pm 11.2$ | $76.0 \pm 3.4$ |
| antmaze-medium-play-v0 | 81.3 | 76.6 | 83.6 | 84.5 | 80.7 | $84.6 \pm 4.2$ | $62.5 \pm 3.7$ |
| antmaze-medium-diverse-v0 | 82.0 | 78.6 | 83.8 | 84.8 | 75.0 | $69.0 \pm 5.6$ | $67.2 \pm 2.0$ |
| antmaze-large-play-v0 | 59.3 | 46.4 | 66.6 | 63.5 | 53.6 | $49.2 \pm 11.2$ | $40.1 \pm 8.6$ |
| antmaze-large-diverse-v0 | 64.8 | 56.6 | 64.8 | 67.9 | 53.6 | $69.4 \pm 15.2$ | $60.5 \pm 3.7$ |
| **Average (AntMaze)** | 74.2 | 69.6 | 78.3 | 79.1 | 73.6 | $73.8 \pm 8.0$ | $66.5 \pm 3.8$ |

We observe that GMPG's performance in the AntMaze environment does not surpass that of GMPO, unlike in the locomotion medium and medium-replay tasks. This is particularly evident in the AntMaze medium and large tasks, where the maze size increases and GMPG becomes less effective. This discrepancy may result from the increased complexity and task length of the AntMaze environment, which demands stable and effective policy generation. The goal in this environment is timely navigation through the maze, not rapid completion; faster trajectories do not yield higher rewards. Aggressive strategies can cause ant agents to fall and fail to complete the task, potentially explaining the performance difference between GMPO and GMPG in AntMaze.

## D.3  ABLATION EXPERIMENTS

**Temperature Coefficient $\beta$**    The temperature coefficient $\beta$ is a common hyperparameter in both GMPO and GMPG. Larger $\beta$ values indicate stronger exploration. As shown in Table 12, a moderate $\beta$ value is beneficial for performance.

Table 12: Performance comparison of temperature coefficient $\beta$ for GMPO and GMPG.

| Algo. / Model type / Function Type Pretrain scheme / Fintune scheme | GMPG / GVP / $v(x_t,t)$ - / Eq. 40 | | | GMPG / VPSDE / $v(x_t,t)$ Eq. 3 / Eq. 8 | | |
|---|---|---|---|---|---|---|
| $\beta$ | 1 | 4 | 8 | 1 | 4 | 8 |
| halfcheetah-medium-v2 | $43.2 \pm 1.2$ | $48.9 \pm 1.9$ | $49.9 \pm 2.7$ | $57.0 \pm 3.1$ | $56.1 \pm 2.7$ | $55.5 \pm 2.6$ |
| halfcheetah-medium-replay-v2 | $40.6 \pm 2.9$ | $42.3 \pm 3.6$ | $42.2 \pm 3.1$ | $47.1 \pm 2.5$ | $50.5 \pm 2.7$ | $50.0 \pm 3.0$ |

**Solver Schemes for Sampling.** Table 13 shows the performance comparison of different solver schemes for GMPO and GMPG. We find GMPO and GMPG to be robust to the choice of solver schemes, with performance remaining consistent across different schemes.

Table 13: Performance comparison using different solver schemes for GMPO/GMPG. RK4 stands for the Fourth-order Runge-Kutta with $3/8$ rule. DPM-Solver of order 2 is used for 17 steps sampling. Other solver schemes are used for 32 steps sampling. The average performance is very close across different solver schemes.

| Pretrain scheme / Finetune scheme | GMPO / VPSDE / $v(x_t, t)$ - / Eq. 40 | | | |
|---|---|---|---|---|
| Solver schemes | Euler-Maruyama | Midpoint | RK4 | DPM-Solver |
| hopper-medium-v2 | $78.5 \pm 20.2$ | $76.9 \pm 20.4$ | $75.0 \pm 22.1$ | $76.7 \pm 19.2$ |
| halfcheetah-medium-expert-v2 | $89.6 \pm 4.3$ | $83.5 \pm 8.9$ | $83.6 \pm 4.7$ | $90.9 \pm 3.6$ |
| Pretrain scheme / Finetune scheme | GMPG / VPSDE / $v(x_t, t)$ Eq. 3 / Eq. 8 | | | |
| Solver schemes | Euler-Maruyama | Midpoint | RK4 | DPM-Solver |
| hopper-medium-v2 | $101.1 \pm 2.6$ | $98.3 \pm 9.6$ | $98.9 \pm 2.1$ | $100.5 \pm 2.2$ |
| halfcheetah-medium-expert-v2 | $89.0 \pm 6.4$ | $88.2 \pm 5.4$ | $88.2 \pm 4.3$ | $89.7 \pm 4.0$ |

# E  FRAMEWORK

*GenerativeRL* is implemented using native *PyTorch 2.0* and utilizes numerical solvers like *torchdiffeq* (Chen et al., 2018) and *Torchdyn* (Poli et al., 2021) for ordinary differential equations. Unlike frameworks such as Hugging Face Diffusers (von Platen et al., 2022) that generate final outputs, in reinforcement learning (RL), generative models act as policies, world models, or other components. This requires a differentiable sampling process with computable and differentiable likelihoods. *GenerativeRL* addresses this by supporting sampling $x \sim p(\cdot|c)$ and computing $\log p(x|c)$ with or without automatic differentiation, uniformly across all continuous-time generative models.

## E.1  USAGE EXAMPLE

Figure 4 illustrates a usage example of *GenerativeRL*. The framework is user-friendly and designed for ease of use by reinforcement learning researchers. All models, training, and sampling processes are defined in a single script, simplifying understanding and modifications. Experiment configurations are recorded automatically, facilitating result reproduction.

Modular settings allow users to switch between different generative models, neural network components, and special configurations easily. Once the model is defined, training and sampling require only a few lines of code. Users can switch between training objectives and inference strategies seamlessly. Most functions for generative models support batch processing and automatic differentiation, ensuring smooth integration into reinforcement learning algorithms.

## E.2  FRAMEWORK STRUCTURE

The framework structure is illustrated in Figure 5. It consists of three main components: reinforcement learning algorithms, generative models, and neural network components. The reinforcement learning algorithms include those based on generative models, such as QGPO and SRPO. The generative models, including diffusion, flow, and bridge models, are used to model data distributions and serve as key components in RL algorithms. The neural network components comprise commonly used layers like DiT and U-Net, which are essential for building generative models. Users can customize the neural network components and generative models to fit their specific needs. The framework is designed to be compatible with future RL algorithms and can be easily extended for different RL tasks.

The currently supported generative models are listed in Table 14.

Table 14: Models functionality in *GenerativeRL*

| Name | Score Matching | Flow Matching | $x \sim p(\cdot)$ | $x \sim p(\cdot\|c)$ |
|---|---|---|---|---|
| Diffusion models | | | | |
| VPSDE | ✓ | ✓ | ✓ | ✓ |
| GVP | ✓ | ✓ | ✓ | ✓ |
| Linear (Karras et al., 2022) | ✓ | ✓ | ✓ | ✓ |
| Flow models | | | | |
| I-CFM | ✗ | ✓ | ✓ | ✓ |
| OT-CFM (Tong et al., 2024) | ✗ | ✓ | ✓ | ✗ |
| Bridge models | | | | |
| SF2M (Tong et al., 2023) | ✗ | ✗ | ✓ | ✗ |

✓ Supported. ✗ Not supported.

## F  LIMITATIONS AND FUTURE WORK

Our work has several limitations:

- We focused on policy extraction using an optimal Q-value function trained with IQL, without considering suboptimal Q-value functions or improving Q-value estimation using generative models. This remains an open question for actor-critic methods utilizing generative models as policies.

- Generative models excel in high-dimensional data generation but may perform similarly to discriminative models in low-dimensional contexts. Since most RL environments are low-dimensional, future work should explore environments like Evogym (Bhatia et al., 2021), where generative policies can leverage larger action spaces.

- We did not address scenarios where generative policies are deployed online with real-time data generation. Future studies should assess the stability of policy optimization in dynamic data contexts, evaluate performance, and consider the additional costs of using generative policies.

## G  SOCIETAL IMPACTS

This work uses generative models as policy models in reinforcement learning, applicable in real-world applications such as robots and autonomous vehicles. Given uncertainties in Q-value function training and reward labeling, potential negative societal impacts exist, including the risk of aligning generative models with harmful Q-value functions. Therefore, caution is essential throughout the training and deployment process, from data collection and reward labeling to Q-value function training and the application of generative policies. Additionally, we must prevent the abuse of this technology, such as using it for illegal activities or unethical purposes.

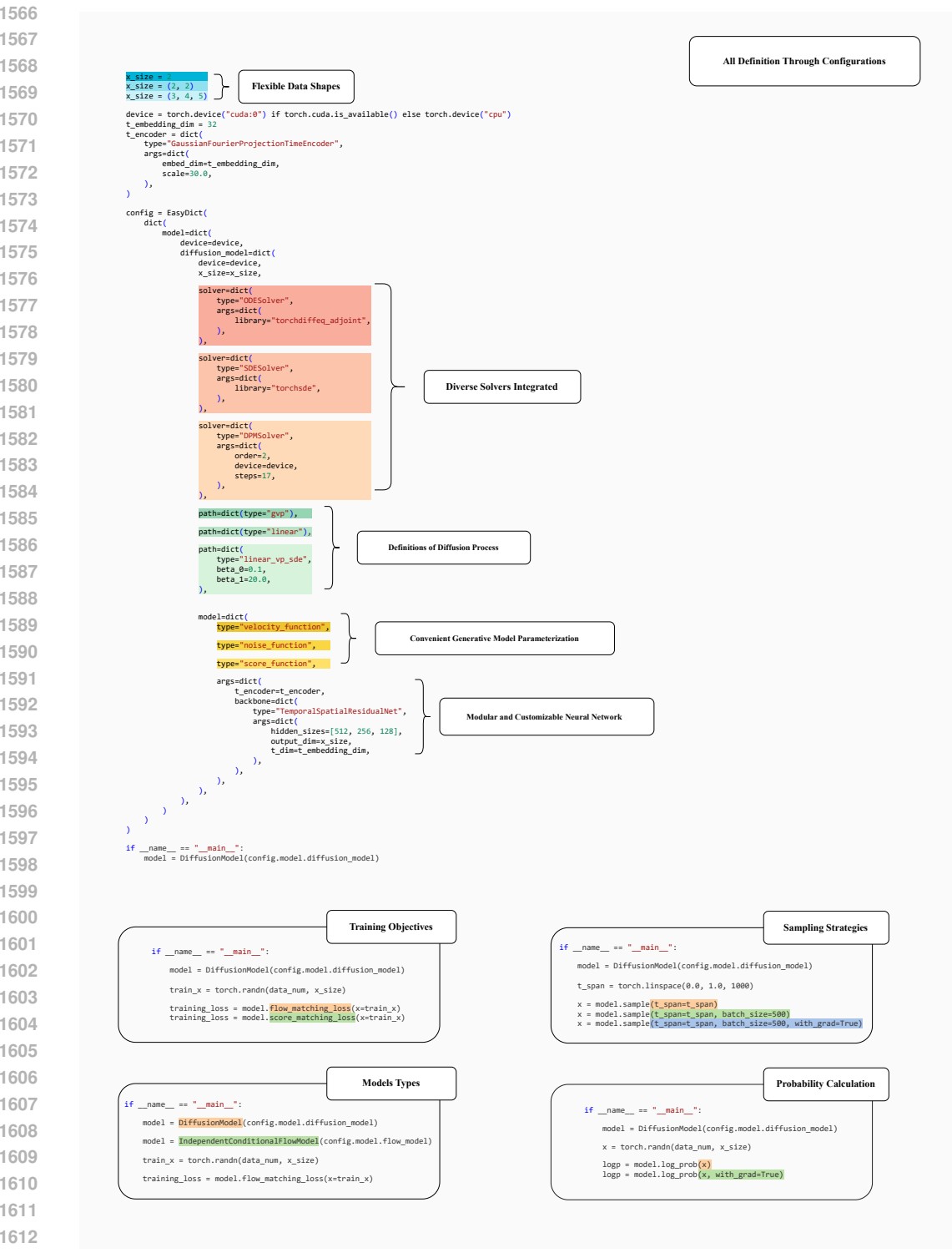

Figure 4: An example of using *GenerativeRL* for defining models, training, and sampling. All experiment configurations are orgnized in a nested dictionary and can be recorded for reproductions. Configuration of every component is modular and can be easily switched. Diverse generative models and neural network components are supported. User can switch between training objectives and inference strategies easily. Most functions support batch processing and automatic differentiation with only a few lines of code. This configuration is flexible and can be easily extended for different RL tasks.

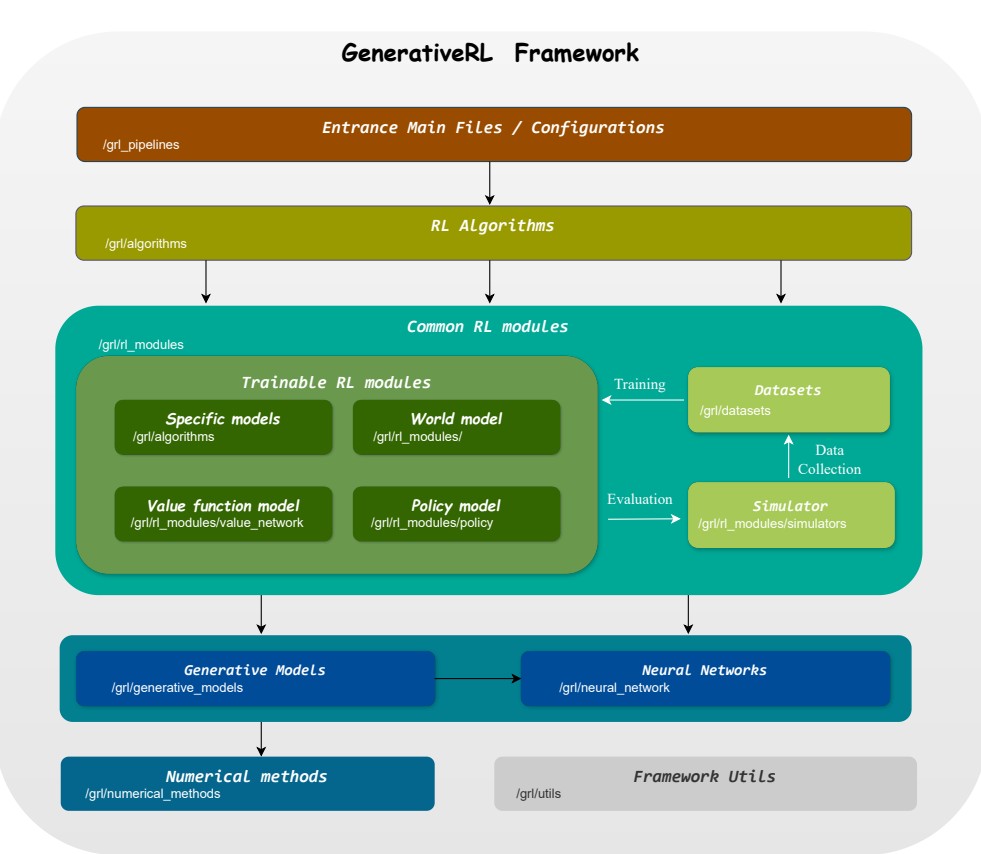

Figure 5: Framework structure of *GenerativeRL*.

