# OpenReview forum: "Revisiting Generative Policies: A Simpler Reinforcement Learning Algorithmic Perspective"
_ICLR.cc/2025/Conference — Submitted to ICLR 2025_

### Official Review · Reviewer_ioaC · 2024-10-24

**Soundness:** 3
**Presentation:** 1
**Contribution:** 2
**Rating:** 5
**Confidence:** 4

**Summary:**

This paper revisits prior research on generative policies and provides a comprehensive comparison and analysis of various learning objectives and optimization techniques, proposing a simpler approach to reinforcement learning. The paper introduces two key methods: Generative Model Policy Optimization (GMPO) and Generative Model Policy Gradient (GMPG). GMPO leverages a more straightforward advantage-weighted regression method, enabling efficient training, while GMPG introduces a numerically stable policy gradient approach. Additionally, the paper presents an experimental 'software' named GenerativeRL, designed to evaluate the performance of generative policy algorithms and demonstrate their consistent results across multiple datasets. Through this work, the authors propose a novel methodology combining generative models with reinforcement learning, simplifying the training process without compromising performance.

The key concepts covered in the paper are as follows:
1. Generative Policy: A method in reinforcement learning that utilizes generative models to represent the distribution of actions when learning a policy. This is particularly effective in continuous action spaces and is exemplified by diffusion models and flow models, which excel in modeling multi-modal data.
2. GMPO (Generative Model Policy Optimization): A training method that uses advantage-weighted regression to optimize policies. GMPO simplifies the training process, offering stability and efficiency without the need for behavior policy, while extracting optimal policies based on environmental rewards.
3. GMPG (Generative Model Policy Gradient): A variant of the policy gradient method that provides a numerically stable implementation. Unlike GMPO, GMPG builds on pretrained policies and computes gradients over continuous time.
4. GenerativeRL Framework: A standardized experimental framework designed to evaluate and compare different generative policy learning algorithms. It decouples the generative model from the reinforcement learning components, providing unified API, flexible data formats and shapes, auto-grad support, and diverse generative model integration.

**Strengths:**

- GMPO and GMPG effectively simplify complex training methods, operating efficiently while delivering optimal policies without performance degradation.
- The proposed methods demonstrate comparable when compared to state-of-the-art algorithms across various datasets, consistently proving their stability and robustness in different settings.
- GenerativeRL Framework: The detailed description of the framework provided in the appendix significantly enhances reproducibility, which is a strong point. (However, I have some reservations regarding the extent of the contribution attributed to this framework, which will be discussed in the weaknesses section.)

**Weaknesses:**

I raise several concerns about the overall structure of this paper.
- Firstly, it is unclear whether the authors are proposing a "framework" or merely organizing "experimental code" more systematically.
Secondly, it remains ambiguous whether the paper "classifies existing works into two categories(line 18 in abstract)" or surveys prior works, "identifies their limitations, and proposes two new algorithms" as a solution.

    - 3.1: The term "experimental framework" should be explicitly defined as referring to software. While the well-organized experimental code is appreciated, I question whether emphasizing it as a "framework" and a "contribution" is justified.

    - 3.2: The connection between GMPO, GMPG, and existing algorithms is not sufficiently clear. I also find the theoretical foundation of these algorithms lacking. Specifically:
        1. The introduction does not provide a thorough explanation of why these categories were chosen or how they were grouped. A more explicit revisit is needed. For example, if SfBC belongs to the GMPx class, and QGPO is also part of the GMPx class, it should be clarified what their common traits are, and why they were categorized together. Then, the paper could explain how the proposed GMPO and GMPG represent advances based on these categories.
        2. In Section 4.2: Previous Works on Generative Policy, while it is clear what improvements are needed, there is no clear categorization presented. Additionally, while systematically investigating inefficiencies and eliminating them is always a valuable academic contribution, there is no theoretical proof supporting the proposed methods. The paper gives reasons why the methods might be necessary, but it does not clearly show how they improve the situation.

- It is unclear whether this study focuses specifically on offline RL or is meant to be more broadly applicable to general RL.

- In the Related Works section, the connection between prior research and the present work is not sufficiently emphasized. The section reads more like a chronological listing without interpretation of the strengths, weaknesses, or the progression of improvements. It would also be helpful to introduce abbreviations (e.g., SfBC) in the Related Works section and explain the paper’s categorization in that context.

**Questions:**

- What does "Any" in Table 1 signify? If it only refers to DDPM and VPSDE, does that mean methods based on flow models are not covered? Or, are flow models also included under "Any"? (I understand "Any" includes flow but it confuse me.) Were there no prior algorithms using flow models, or were they intentionally excluded? In the introduction, it was mentioned that offline RL focuses on diffusion models—are there no flow-based methods in this domain? (line 44, Table 1)

- Can the proposed GMPO and GMPG methods be stably applied to more complex generative models beyond diffusion and flow models?

- Are there any reinforcement learning applications using generative policies that are not focused on offline RL? The scope of the research remains unclear to me.

- How well does the flexibility of the GenerativeRL framework extend to the compatibility with other reinforcement learning algorithms?

- The abbreviations DSM and CFM do not appear to be explained in "the front" of the paper (eqn 2,3). Could these be clarified?

---

> ### Author Response · Authors · 2024-11-19
>
> Thanks for your feedback and all suggestions! I'm glad you appreciated our work.
>
> Here are some answers to your questions:
>
> ---
>
> > 'Are the authors proposing a "framework" or merely organizing "experimental code"?'
> > 'How well does the flexibility of the GenerativeRL framework extend to the compatibility with other reinforcement learning algorithms?'
>
> GenerativeRL is a framework designed for research at the intersection of generative models and reinforcement learning (RL). It goes beyond just organizing experimental code, offering key features that distinguish it as a framework:
>
> **Key Features of GenerativeRL**:
>
> **Modularity**: GenerativeRL is highly modular, allowing users to swap or combine different components, such as generative models or RL algorithms. This flexibility is demonstrated across the experiments in the paper and further detailed in the appendix.
>
> **Extensibility**: The framework provides a clear API for adding or modifying components. Users can customize neural network architectures, loss functions, and training procedures while reusing the provided generative models and RL algorithms. For instance, one can easily integrate a new diffusion model by modifying only the diffusion process coefficients, leaving the rest unchanged. Additionally, the generative models and RL algorithms can be used independently, allowing users to integrate their own datasets or environments.
>
> **Compatibility**: GenerativeRL was designed with long-term maintenance and development in mind. It defines flexible data shapes and types, as detailed in the Framework section. Additionally, it supports automatic differentiation, which is crucial for deep learning, and is compatible with distributed training using either native PyTorch DDP or libraries like Accelerate.
>
> **Minimal Dependencies**: We’ve taken a minimalistic approach to dependencies, focusing on clarity and simplicity.
>
> **Unittests**: The codebase includes extensive unittests to ensure the correctness of the implementation. These tests cover the core components of the framework, such as generative models and RL modules, running automatically during continuous integration.
>
> **Documentation**: The codebase is well-documented, with detailed explanations of how components interact. It includes guides on implementing generative models and RL algorithms, with predefined data shapes, class types, and function augmentations. We provide Jupyter notebook examples and toy demos to help users get started quickly.
>
> **Open-source**: GenerativeRL is designed for open-source, allowing the community to contribute and extend the framework. We welcome feedback, bug reports, and pull requests to improve the framework and make it more accessible to researchers and practitioners via PYPI.
>
> We apologize that we cannot provide more information in the paper and supplementary materials due to review anonymity rules, but we have provided most of the ready-made code that constitutes this open source library in the supplementary materials.
>
> We are committed to maintaining and improving GenerativeRL by integrating new research and community feedback. We hope that the framework will serve as a valuable tool for researchers and practitioners working on generative models and RL.
>
> ---
>
> > 'Does the paper "classify existing works into two categories" or "survey prior works, identify their limitations, and propose two new algorithms as a solution"?'
>
> The paper does both: it classifies existing works into two categories and surveys prior works, identifying their limitations and proposing two simplified algorithms as a solution.
>
> By comparing existing methods, we aim to provide a comprehensive overview of the state-of-the-art in generative policies. Rather than introducing new techniques in GMPO or GMPG, we simplify existing approaches into two forms. This simplification is motivated by our observation that many methods share similar training objectives, despite being implemented differently.
>
> We categorize these methods based on two core objectives:
>
> - Maximizing an advantage-weighted log-likelihood.
> - Maximizing a guided function sampled from the generative model, with proximal constraints.
>
> From a foundational perspective, we argue that these two objectives capture the essence of current methods, with the rest being implementation details. Many existing methods are overly complex and obscure which components are crucial for performance. For instance, techniques like action tensor clipping in works such as Diffusion-QL are rarely discussed explicitly.
>
> Additionally, most existing methods lack modularity, making it difficult to swap components for ablation studies or comparison.
>
> Our experiments demonstrate that the proposed algorithms are competitive with existing methods, while being simpler and less technically decorated.

---

> ### Author Response · Authors · 2024-11-19
>
> ---
>
> > 'Does this study focus specifically on offline RL, or is it broadly applicable to general RL?'
>
> The experiments in this paper focus on offline RL, but the proposed methods, GMPO and GMPG, are general and can be applied to online RL as well.
>
> We conducted some preliminary experiments using a two-stage training process: first, we pretrained the generative policy with offline datasets, then transitioned to online RL, where new data was collected and used to fine-tune both the policy and value function. These experiments showed that the online fine-tuned policy outperformed the offline pretrained one, demonstrating the methods' competitiveness in online RL. Similar experimental results can be found in Appendix F of the aforementioned work Diffusion-QL's paper, where they also found that using online data would further improve model performance.
>
> However, since our focus in this paper is on revisiting generative policy optimization in offline RL, we leave a more systematic exploration of online RL for future work. In general, online RL poses additional challenges, such as balancing exploration and exploitation, which are beyond the scope of this paper.
>
> ---
>
> > 'The connection between prior research and the present work is not sufficiently emphasized. It would also be helpful to introduce abbreviations (e.g., SfBC) in the Related Works section and explain the paper’s categorization in that context.'
>
> Thank you for the suggestion. We will revise the paper to better emphasize the connection between our work and prior research, including the use of abbreviations in the Related Works section.
>
> We already provide a brief overview of these connections in Appendix C, but we will ensure this information is included in the main text for clarity. We apologize for omitting it due to page limitations.
>
> ---
>
> > 'What does "Any" in Table 1 signify? If it only refers to DDPM and VPSDE, does that mean methods based on flow models are not covered? Or, are flow models also included under "Any"?'
>
> The term "Any" in Table 1 refers to the inclusion of all generative modeling methods, encompassing both diffusion models (such as DDPM and VPSDE) and flow-based models. Flow models are also considered under the "Any" category. To clarify this further, we will update Table 1 with additional details to avoid any potential misunderstandings.
>
> ---
>
> > 'Were there no prior algorithms using flow models, or were they intentionally excluded? In the introduction, it was mentioned that offline RL focuses on diffusion models—are there no flow-based methods in this domain? (line 44, Table 1)'
>
> Flow-based methods do exist in the domain of reinforcement learning (RL), and there is growing interest in using flow models for decision-making tasks. Since the introduction of flow matching for generative models in late 2022, over 500 papers have cited this approach, with more than 90 papers mentioning reinforcement learning (RL). Some of these works either introduce new RL algorithms or adapt flow-based methods to RL settings.
>
> Here are several notable examples of flow-based methods in RL and decision-making:
> - ["Preference Alignment with Flow Matching"](https://arxiv.org/abs/2405.19806): This work explores preference-based reinforcement learning (PbRL) using flow matching, similar to diffusion-based approaches but with flow models.
> - ["RF-POLICY: Rectified Flows are Computation-Adaptive Decision Makers"](https://openreview.net/forum?id=hQERBmmlYm): This paper introduces an imitation learning algorithm that utilizes rectified flows for adaptive decision-making.
> - ["Guided Flows for Generative Modeling and Decision Making"](https://arxiv.org/abs/2311.13443): This work presents guided flows for generative modeling, with experiments demonstrating their application in reinforcement learning.
> - ["Riemannian Flow Matching Policy for Robot Motion Learning"](https://arxiv.org/abs/2403.10672): A method focused on robot motion learning, integrating flow-based generative policies with Riemannian manifold constraints.
> - ["Affordance-based Robot Manipulation with Flow Matching"](https://arxiv.org/abs/2409.01083): This paper applies flow-based models to robot manipulation tasks, driven by text-prompted affordance learning.
> - ["Generalized Schrödinger Bridge Matching"](https://arxiv.org/abs/2310.02233): A generalized approach for bridging distribution matching in reinforcement learning, leveraging a flow-based framework.
> - ["π0: A Vision-Language-Action Flow Model for General Robot Control"](https://arxiv.org/abs/2410.24164): This work applies flow models in vision-language-action tasks for robot control, demonstrating the versatility of flow-based policies.
>
> Although these papers reflect the growing use of flow models in RL, they were not included in our current comparisons, mainly due to the particular focus of our work is different from these papers.
>
> We will consider adding these references to provide a more comprehensive overview.

---

> > ### Author Response · Authors · 2024-11-19
> >
> > ---
> >
> > > 'Can the proposed GMPO and GMPG methods be stably applied to more complex generative models beyond diffusion and flow models?'
> >
> > Yes, the proposed GMPO (Generative Model Policy Optimization) and GMPG (Generative Model Policy Gradient) methods can be stably applied to more complex generative models beyond diffusion and flow models.
> >
> > **GMPO**: This method can be viewed as an advantage-weighted log-likelihood maximization approach, making it a general objective that can be applied to a wide range of generative models. If a generative model is trainable with a likelihood-based objective, GMPO can be used for optimizing generative policies by maximizing the model's log-likelihood while incorporating the advantage function. This allows GMPO to extend to more complex models, such as autoregressive models, as long as they involve likelihood-based training.
> >
> > **GMPG**: This method leverages policy gradient optimization combined with a guided function applied to samples drawn from the generative model, with proximal constraints to ensure stable learning. As long as the generative model supports automatic differentiation and the guided function can be computed via automatic differentiation, GMPG can be applied directly. Even in scenarios where the guided function is not differentiable, GMPG can be adapted into a REINFORCE-like algorithm, which uses gradient estimation techniques to optimize the generative policy.
> >
> > In summary, both GMPO and GMPG are flexible and general enough to be applied to a wide variety of generative models, extending beyond diffusion and flow-based models, provided the models meet certain conditions like likelihood-based training or differentiability for sampling and guiding functions.
> >
> > ---
> >
> > > 'Are there any reinforcement learning applications using generative policies that are not focused on offline RL? The scope of the research remains unclear to me.'
> >
> > Yes, generative policies are applied beyond offline reinforcement learning (RL). One prominent example is in embodied AI and robotics, where generative models are used to control robots in real-time environments.
> >
> > A recent work that exemplifies this trend is ["RDT-1B: a Diffusion Foundation Model for Bimanual Manipulation"](https://rdt-robotics.github.io/rdt-robotics/), which uses a diffusion-based generative model for controlling robotic manipulators. This model supports both zero-shot generalization and few-shot learning, which are crucial for adaptive control in robotics. Such applications demonstrate that generative policies, while helpful in offline RL, are also valuable in real-time decision-making tasks in robotics and other interactive systems.
> >
> > Our work will help to more deeply elucidate the training mechanism of generative policies in reinforcement learning and finally benefit a broader range of these applications.
> >
> > ---
> >
> > > 'The abbreviations DSM and CFM do not appear to be explained in "the front" of the paper (eqn 2,3). Could these be clarified?'
> >
> > Yes, the abbreviations DSM and CFM stand for:
> > - DSM: "Denoise Score Matching," a method used in diffusion models to learn the score function of data distributions by denoising corrupted samples.
> > - CFM: "Conditional Flow Matching," a technique used in flow-based generative models, focusing on matching conditional distributions through flow transformations.
> >
> > These abbreviations are clarified in Appendix B.1 and B.2 of the paper.
> > These terminologies are introduced from previous works.
> > - DSM originates from prior work on denoising score matching, specifically from the paper titled ["A connection between score matching and denoising autoencoders"](https://www.iro.umontreal.ca/~vincentp/Publications/smdae_techreport.pdf).
> > - CFM is based on prior advancements in flow-based generative models, detailed in ["Improving and generalizing flow-based generative models with minibatch optimal transport"](https://arxiv.org/abs/2302.00482).

---

> ### Comment · Reviewer_ioaC · 2024-11-25
> **Response to Rebuttal**
>
> Dear authors, thanks for the rebuttal. I have carefully read your response and some of concerns are resolved. Currently, I have a follow-up question:
> * In the revised manuscript, I have not able to clearly identify a formal derivation establishing theoretical connection between two objectives: "maximizing an advantage-weighted log-likelihood" and  "maximizing a guided function sampled from the generative model". Could you elaborate or kindly point this connection in a rigorous way?

---

> > ### Author Response · Authors · 2024-11-25
> >
> > Dear reviewer,
> >
> > Thank you for your valuable suggestions and for pointing out these concerns.
> >
> > We have added the relevant explanations in Appendix A.2 of the latest version of the paper (PDF) to illustrate the connection and the differences between these two methods. We hope this addresses your concerns.

---

> > > ### Comment · Reviewer_ioaC · 2024-11-25
> > > **Thank you**
> > >
> > > Dear authors, I appreciate your response. From our discussion, I raised the score from 3 to 5. I will finalize my score after discussion with other reviewers.

---

### Official Review · Reviewer_zQTz · 2024-10-24

**Soundness:** 3
**Presentation:** 4
**Contribution:** 3
**Rating:** 8
**Confidence:** 4

**Summary:**

This paper 1. summarizes previous optimization methods for diffusion policies and classifies them into two categories. 2. proposes a simpler optimization method for each optimization category. The paper also abstracts existing diffusion optimization techniques into a Python library and (maybe) plans to open-source it. The above algorithms are compared in detail in standard benchmarks.

**Strengths:**

1. This is the first work I know that integrates various diffusion policy optimization algorithms into one framework. I believe this is beneficial to the community.
2. The work presents a clear theoretical comparison of existing diffusion optimization methods and unifies them in one framework. This is good contribution.
3. The proposed GMPO and GMPG algorithm is theoretically foundationed and yields good performance compared with previous work.
4. The paper is well-presented. Table 1 and Table 2 is extremely helpful in understanding current lines of work.

**Weaknesses:**

1. I have some concerns about the novelty of GMPO. Despite its simplicity, I believe this is somewhat very similar to the AWR algorithm, just that it optimizes a diffusion policy instead of a Gaussian policy. Also, this algorithm has already been verified in the image generation field.

Kimin Lee, Hao Liu, Moonkyung Ryu, Olivia Watkins, Yuqing Du, Craig Boutilier, Pieter Abbeel, Mohammad Ghavamzadeh, and Shixiang Shane Gu. Aligning text-to-image models using human feedback. arXiv preprint arXiv:2302.12192, 2023. 3, 6

2. When applying GMPG, as I understand, Neural ODE needs to be leveraged in order to estimate logp of \mu. I'm concerned about the computation efficiency of this method. Could you compare it with SRPO and Diffusion-QL? Also, I think for GMPG, your policy model has to be a Gaussian but cannot be a diffusion policy. For Diffusion QL, your policy can be a Diffusion policy. Is this correct?

3. There are mistakes in the inference scheme for SfBC, and Diffusion QL, and IDQL in Table 2. I recall for Diffusion QL and IDQL, they simply take [max] of action candidate Q values, not softmax. For SfBC, it uses [softmax] rather than [max]. Not entirely sure about the first two. Please recheck their code implementations.

**Questions:**

1. When will the codebase go public?
2. I suggest annotating each algorithm and indicating which category they come from in Table 2. For instance, Diffusion QL and SRPO are reverse KL methods.   QGPO and GMPO are forward KL. SfBC and IDQL are forward KL (or sampling-based?)  This should be more clear to readers.

---

> ### Author Response · Authors · 2024-11-22
>
> Thanks for your all feedbacks and nice suggestions! I'm glad you appreciated our work.
>
> Here are some responses to your questions:
>
> ---
>
> > 'I have some concerns about the novelty of GMPO. Despite its simplicity, I believe this is somewhat very similar to the AWR algorithm, just that it optimizes a diffusion policy instead of a Gaussian policy. Also, this algorithm has already been verified in the image generation field.'
>
> It’s true that GMPO bears a structural similarity to AWR in a narrow sense, as the latter uses the following form:
>
> $\mathcal{L}\_{\mathrm{AWR}}(x)=\exp{A(x)}\log{p(x)}$
>
> In fact, any generative model that can be trained with a maximum likelihood objective, $\mathcal{L}\_{\mathrm{MLE}}(x)$, to maximize its log-likelihood (without requiring a strict bound), can be applied with the AWR framework in a general sense:
>
> $\mathcal{L}\_{\mathrm{AWR}}(x)=\exp{A(x)}\mathcal{L}\_{\mathrm{MLE}}(x)$
>
> For GMPO, the $\mathcal{L}\_{\mathrm{MLE}}(x)$ corresponds to the flow matching loss or score matching loss. Both of these methods have been shown to serve as evidence lower bounds (ELBOs) for maximum likelihood estimation in diffusion or flow models. This has been formally proven in the following works:
> - ["Maximum likelihood training of score-based diffusion models"](https://arxiv.org/abs/2101.09258)
> - ["Maximum likelihood training for score-based diffusion odes by high order denoising score matching"](https://arxiv.org/abs/2206.08265)
> - ["Improved techniques for maximum likelihood estimation for diffusion odes"](https://arxiv.org/abs/2305.03935).
>
> In this sense, GMPO can be seen as a generalization of AWR, as it extends the framework to diffusion and flow models.
>
> ---
>
> > 'When applying GMPG, as I understand, Neural ODE needs to be leveraged in order to estimate logp of \mu. I'm concerned about the computation efficiency of this method. Could you compare it with SRPO and Diffusion-QL?'
>
> We can compare with SRPO based on our implementation.
>
> For the finetuning stage of halfcheetah-medium-expert-v2-SRPO, it reaches its best performance after approximately 1.8 hours on a single A100 GPU.
> For the finetuning stage of halfcheetah-medium-expert-v2-GMPG, it reaches its best performance after approximately 1.7 hours on a single A100 GPU.
>
> The time may vary across datasets and tasks, but the computational cost of GMPG is not significantly different from SRPO in practice.
>
> ---
>
> > 'Also, I think for GMPG, your policy model has to be a Gaussian but cannot be a diffusion policy. For Diffusion QL, your policy can be a Diffusion policy. Is this correct?'
>
> This is not entirely correct.
>
> For GMPG, the pretrained policy can be either a diffusion model or a flow model, obtained via score matching or flow matching, as long as $\log{\pi}$ is analytical and can be computed.
> After finetuning, the policy is no longer the original pretrained diffusion or flow model—it becomes a flow model with a modified generation trajectory.
> Importantly, $\log{\pi}$ remains analytical for the finetuned policy.
>
> For Diffusion-QL, the policy is a diffusion model with score matching constraints in the original paper. However, it can be extended to a flow model with flow matching constraints as well.
>
> For both GMPG and Diffusion-QL, the policy can degenerate into a Gaussian model, which is equivalent to the DDPG algorithm in a general sense.

---

> > ### Author Response · Authors · 2024-11-22
> >
> > ---
> >
> > > 'There are mistakes in the inference scheme for SfBC, and Diffusion QL, and IDQL in Table 2. I recall for Diffusion QL and IDQL, they simply take [max] of action candidate Q values, not softmax. For SfBC, it uses [softmax] rather than [max]. Not entirely sure about the first two. Please recheck their code implementations.'
> >
> > Thank you for pointing this out. We have rechecked the code implementations and have clarified the inference schemes for SfBC, Diffusion QL, and IDQL, correcting the representations in Table 2.
> >
> > **SfBC**:
> > The inference scheme in SfBC is effectively a max function in a softmax-like form, as the algorithm uses a very large temperature in the exponential function. This can be verified in their [code implementation](https://github.com/ChenDRAG/SfBC/blob/master/utils.py#L60).
> > The inference scheme is also described in their [paper](https://arxiv.org/pdf/2209.14548).
> >
> > - In Section 3.2: "Finally, an action is resampled from M candidates with $\exp(\alpha Q_{\phi}(s, a))$ being the sampling weights. We summarize this procedure as selecting from behavior candidates (SfBC), which could be understood as an analogue to rejection sampling."
> > - In Appendix B.1 (Evaluation): "During algorithm evaluation, we select actions in a deterministic way. Specifically, the action with the highest Q-value within M behavior candidates will be selected for environment inference during evaluation."
> >
> > Your suggestion is better. While the deterministic "max" is used during evaluation, the softmax form (with a large temperature) is more accurate to describe the method.
> >
> > **Diffusion QL**:
> > Although the paper does not explicitly mention this, the [code implementation for evaluation](https://github.com/Zhendong-Wang/Diffusion-Policies-for-Offline-RL/blob/master/main.py#L160) shows that a softmax function is used during inference, as seen in this [line](https://github.com/Zhendong-Wang/Diffusion-Policies-for-Offline-RL/blob/master/agents/ql_diffusion.py#L193). This confirms the use of softmax weighting rather than a simple max function.
> >
> > **IDQL**:
> > The inference scheme is described in Algorithm 2 of the [papr](https://arxiv.org/pdf/2304.10573), where the action is sampled from a categorical distribution formed by the normalized weighting value $w$. This corresponds to a softmax function.
> >
> > Additionally, in Section 4.1, the authors state: "In practice, we also found that simply taking the action with the highest Q-value tends to yield better performance at evaluation time, which corresponds to selecting $w(s,a)$ to be a one-hot. This yields the deterministic policy."
> >
> > Therefore, both interpretations are correct: the deterministic argmax can be seen as a special case of the softmax function when the temperature approaches zero.
> >
> > In summary:
> >
> > - SfBC: Softmax-like with a large temperature (deterministic max during evaluation).
> > - Diffusion QL: Softmax during evaluation.
> > - IDQL: Softmax for sampling but often uses deterministic argmax for better performance.
> >
> > We have updated Table 2 accordingly to reflect these corrected inference schemes.
> >
> > ---
> >
> > > 'When will the codebase go public?'
> >
> > Thank you so much for your support and interest! A preview version of the codebase is now available. We are also continuously updating it to include more algorithms, datasets, and model types, maintaining long-term updates for public users.
> >
> > ---
> >
> > > 'I suggest annotating each algorithm and indicating which category they come from in Table 2. For instance, Diffusion QL and SRPO are reverse KL methods. QGPO and GMPO are forward KL. SfBC and IDQL are forward KL (or sampling-based?) This should be more clear to readers.'
> >
> > Thank you for the suggestion! We agree that adding annotations and categorizing each algorithm would make Table 2 clearer and more informative for readers. We will update the table to include these annotations, specifying whether each algorithm uses forward KL, reverse KL, or sampling-based approaches.

---

> > > ### Comment · Reviewer_zQTz · 2024-11-22
> > >
> > > I thank the authors for their responses and will keep my score. Still, I do agree with some other reviewers that the paper could be improved regarding presentations/clarity. I'm happy to see some changes have been made.

---

### Official Review · Reviewer_FfDE · 2024-10-29

**Soundness:** 3
**Presentation:** 3
**Contribution:** 2
**Rating:** 5
**Confidence:** 2

**Summary:**

The paper presents an unified framework for training flow and diffusion models for offline reinforcement learning algorithms. Authors classify existing training objectives into two categories and linked to simpler approaches, GMPO and GMPG. The paper also proposes a standardized experimental framework for evaluating various generative policies.

**Strengths:**

There are several works which utilize generative models for parametrizing policy for offline RL due to various reasons (multi-modal distribution, controllability, …). I think this paper has a significant contribution on analyzing prior generative policies.

Furthermore, I saw multiple papers in Offline RL reproduce the baselines by their own and the fluctuation of the performance is very large. I think the comprehensive evaluation through GenerativeRL may resolve this issue.

**Weaknesses:**

One of the my major concerns is the position of the paper. It is a bit confused that authors want to build a unified benchmark suite for evaluating various generative policies for RL or propose an algorithm that can be easily applied to train generative policies for RL. For the former, it will be better to include some details on experimental setup and training & evaluation procedure. For the latter, more analysis or stronger experimental results should support the author's claim.

Here are some minor concerns:

- While the paper focus on offline RL algorithms, there are several works which utilize generative policies for online reinforcement learning [1, 2]. It might be better to cover those algorithms or the title should be revised to Offline RL.

- It seems that hyperparameter $\beta$ is heavily tuned for different environment and the range of value is quite large. Could authors verify the details on hyperparameter tuning procedure of proposed approaches?


[1] Psenka, Michael, et al. "Learning a Diffusion Model Policy from Rewards via Q-Score Matching." Forty-first International Conference on Machine Learning.

[2] Jain, Vineet, Tara Akhound-Sadegh, and Siamak Ravanbakhsh. "Sampling from Energy-based Policies using Diffusion." arXiv preprint arXiv:2410.01312 (2024).

**Questions:**

In D4RL benchmark, Diffusion-QL achieves state-of-the-art performance even though other baselines are more recently proposed. Could authors validate why such phenomena happens?

In figure 1, what is the optimal policy in this plot exactly? Does it indicate optimal policy introduced in Eq (5)? Is it tractable?

---

> ### Author Response · Authors · 2024-11-20
>
> Thanks for your feedback and suggestions! I'm glad you appreciated our work, which I hope could be useful for your research.
>
> Here are some responses to your questions:
>
> ---
>
> > 'One of the my major concerns is the position of the paper. It is a bit confused that authors want to build a unified benchmark suite for evaluating various generative policies for RL or propose an algorithm that can be easily applied to train generative policies for RL. For the former, it will be better to include some details on experimental setup and training & evaluation procedure. For the latter, more analysis or stronger experimental results should support the author's claim.'
>
> Thank you for this observation. We can understand your concern about the paper's positioning.
>
> The motivation behind GenerativeRL is the inconsistency in how diffusion policies are defined and implemented across the field. Different training schemes, evaluation metrics, and neural architectures make it difficult to compare generative policies objectively. Our contribution, GMPO and GMPG, aims to simplify these variables and retain only the essential components. This allows for a more direct comparison of generative policies.
>
> Our intention, therefore, is not to compete with existing policies or prove superiority but to provide a reliable baseline for future research. The experiments show that our approach performs well in RL tasks, and we believe this can help reset comparisons on a more level foundation. By maintaining this framework, we hope to support future work in this rapidly evolving area.
>
> We hope this clarification can help you better understand our work.
>
> ---
>
> > 'While the paper focus on offline RL algorithms, there are several works which utilize generative policies for online reinforcement learning [1, 2]. It might be better to cover those algorithms or the title should be revised to Offline RL.'
>
> Thank you for these references. We will evaluate them and consider including them into the framework for future comparisons for online RL. We are also exploring online RL with generative models and have observed promising results in offline pretraining followed by online finetuning. A more comprehensive analysis will be provided in a future paper.
>
> ---
>
> > 'In D4RL benchmark, Diffusion-QL achieves state-of-the-art performance even though other baselines are more recently proposed. Could authors validate why such phenomena happens?'
>
> As shown in Table 3, Diffusion-QL performs similarly to other baselines with high-quality medium-expert data. The difference arises with medium and medium-replay data, where Diffusion-QL outperforms others. This suggests that Diffusion-QL is better at learning from lower-quality data.
>
> We observe similar behavior in GMPG, which uses a reverse-KL objective like Diffusion-QL. This objective tends to explore more aggressively, sometimes pushing the policy towards risky regions, which can yield better performance when learning from imperfect data. By contrast, GMPO, QGPO, and IDQL use a forward-KL objective, which acts more conservatively and struggles with lower-quality data.
>
> Diffusion-QL also incorporates several specific design choices—some mentioned in their paper and others in their code—which may contribute to its superior performance:
>
> - **Joint looping training**: Policy and Q-function are trained simultaneously, akin to a moving average, which may stabilize training.
> While GMPG trains Q-function first via IQL, then finetuning policy.
> - **Value function training**: They use a pure argmax Bellman backup, unlike GMPG's IQL training or QGPO's softmax-based approach.
> - **Action clipping**: Diffusion-QL uses action clipping through out the inference from the diffusion policy if the $x_t$ is out of the action space $[-1, 1]$. This introduces a strong human-made prior by clipping actions to the valid range, effectively regularizing the policy. As the Q function is trained with the action sampled from the diffusion policy, it is regularized as well.
> - **Inference Enhancing**: They generate multiple actions during inference and select the highest-value action, which boosts performance slightly.
>
> GMPG does not use such tricks in implementation for providing a pure, simple and naive diffusion policy from the first principle.
>
> ---
>
> > 'In figure 1, what is the optimal policy in this plot exactly? Does it indicate optimal policy introduced in Eq (5)? Is it tractable?'
>
> In Figure 1, the star represents the best-performing policy we obtained during GMPO and GMPG training. It is not the exact optimal policy from Eq (5), but a model checkpoint saved during training. Therefore, it is tractable and represents the best policy we observed empirically.

---

> ### Comment · Reviewer_FfDE · 2024-12-01
>
> Sorry for my late response. While I strongly agree that different training schemes, evaluation metrics, and neural architectures for diffusion policies hinders fair comparison, several tricks suggested by prior works in implementation level is also quite important. For example, authors claim that GMPO and GMPG achieves competitive performance compared to Diffusion-QL without several tricks such as joint training, generate multiple actions and choose one action, etc. However, I think those tricks should not be ignored and if those tricks lead to significant improvement, we should take it.
>
> I appreciated authors to provide a comprehensive review of prior diffusion policies as it is hard to tracki so many papers which are proposed simultaneously nowdays. However, I'm skeptical on future researchers start their research upon this foundations rather than relying on several tricks already proposed in prior papers. Therefore, I keep maintaining my score.
>
> +) I also briefly check the colab notebook demo in the GenerativeRL code and find that it does not work... Could authors check whether there is no bug in the current code?

---

> ### Author Response · Authors · 2024-12-01
>
> Dear Reviewer,
>
> Thank you for your feedback and for pointing out the error parameter in our demo Colab.
>
> We have corrected the configuration issue in the Colab notebook [GMPO](https://colab.research.google.com/drive/1A79ueOdLvTfrytjOPyfxb6zSKXi1aePv#).
>
> We truly appreciate your valuable contribution to GenerativeRL.
>
> Here are some responses to your further questions:
>
> ---
>
> > "However, I think those tricks should not be ignored, and if those tricks lead to significant improvement, we should take it."
>
> We completely agree with your perspective. From an engineering or deployment standpoint, we should adopt every possible trick to maximize the model's performance.
>
> However, for academic exploration, as emphasized in our paper titled *"Revisiting Generative Policies: A Simpler Reinforcement Learning Algorithmic Perspective"*, our primary objective is to investigate the performance improvements that can be achieved by using generative policies with as few additional tricks as possible.
> By doing so, we aim to conduct an empirical study of pure generative policies, rather than diffusion policies enhanced by other techniques. This approach addresses an important gap in prior works.
>
> For example, in Diffusion-QL, neither action clipping nor inference enhancing tricks were explicitly discussed in their paper. This omission might confuse future researchers and could misleadingly attribute the contributions of **action clipping** and **inference enhancing** to the **diffusion policy** itself. This would be problematic, as these tricks are universal and applicable regardless of whether the policy is generative or not.
>
> If other works, such as QGPO, do not apply such tricks, it would not represent a fair ablation to directly compare the performance of QGPO and Diffusion-QL. Our work aims to provide an intermediate and unified comparison of these algorithms in a fairer and more transparent manner.
>
> We hope this addresses your concerns. Thank you again for your insightful comments.

---

### Official Review · Reviewer_4kSb · 2024-10-29

**Soundness:** 3
**Presentation:** 3
**Contribution:** 2
**Rating:** 5
**Confidence:** 4

**Summary:**

This work systematically analyzes existing diffusion-based offline reinforcement learning methods, including their training and inferring strategy. In summary, this work summarizes existing methods into two classes: Generative Model Policy Optimization (GMPO) and Generative Model Policy Gradient (GMPG), which mainly consider the forward KL and inverse KL respectively. In detail, GMPO directly considers advantage-weighted regression formulation and GMPG offers a numerically stable implementation of the native policy gradient method. This work also provides a standardized experimental framework named GenerativeRL for fair comparison.

**Strengths:**

- Providing a unified perspective is important for the community.

- This work has provided detailed codes and instructions. And the standardized experimental framework is meaningful for comparing different algorithms.

- This work consider more general generative models, including diffusion models and flow models.

**Weaknesses:**

- My major concern is about the novelty of this work. Analyzing the forward KL/ inverse KL seems to be a consensus in the community. As the authors have mentioned, lots of previous work has discussed these two formulations, so are any new theoretical insights or empirical findings that emerged from their systematic analysis that go beyond prior work? Although that, I still appreciate the authors' efforts to provide a unified view and codebase, which is beneficial for further research.

- The performance of GMPO and GMPG seems similar to previous methods in locomotion tasks (table 3) and weaker than baselines in antmaze tasks (table 11, several baselines > 75, while GMPO 73.8 and GMPG 66.5). Why does this phenomenon exist, and is there some specific analysis about it? For example, are there any significant characteristics in locomotion tasks and antmaze tasks that result in the proposed algorithm, especially GMPG, performing poorly in the antmaze tasks?

- How do you calculate $\log \pi$ of the diffusion policies in GMPG? The introduction in lines 316-327 is too brief and the explanation in the appendix should be moved partially to the main text.

**Questions:**

See weaknesses above.

---

> ### Author Response · Authors · 2024-11-20
>
> Thanks for your feedback and suggestions! I'm glad you appreciated our work.
>
> Here are some responses to your questions:
>
> ---
>
> > 'Are any new insights or empirical findings that emerged from their systematic analysis that go beyond prior work?'
>
> From a theoretical perspective, our main contribution is the introduction of two new, simplified training schemes for generative models. By classifying prior methods into two categories, we show that these earlier approaches are special cases of our proposed methods using different training and inference schemes.
>
> Empirically, our methods perform comparably or even better than previous approaches on several benchmark datasets and are more efficient to train. This highlights the effectiveness of both our methods and the components of prior techniques. Additionally, through thorough empirical studies, we demonstrate that several biases identified in prior research can be alleviated with our approach.
>
> Overall, we provide a unified and standardized framework for training generative models in RL tasks, which we hope will serve as a valuable reference for future research.
>
> ---
>
> > 'The performance of GMPO and GMPG seems similar to previous methods in locomotion tasks (table 3) and weaker than baselines in antmaze tasks (table 11, several baselines > 75, while GMPO 73.8 and GMPG 66.5). Why does this phenomenon exist, and is there some specific analysis about it? For example, are there any significant characteristics in locomotion tasks and antmaze tasks that result in the proposed algorithm, especially GMPG, performing poorly in the antmaze tasks?'
>
> Our experiments were not optimized for peak performance on individual tasks but to demonstrate the broad applicability of our methods. We conducted a rough hyper-parameter search using a coarse grid (e.g., geometric sequences with multiples of 2) and selected configurations with generally strong performance. Unlike other works, we did not perform detailed hyper-parameter tuning for each task, which may explain suboptimal results in specific cases. However, overall, our methods' performance falls within the variance observed across these tasks, as we will elaborate in the following analysis.
>
> The antmaze tasks are particularly challenging because the agent must navigate through mazes without getting stuck or falling over. In contrast, locomotion tasks are simpler, typically requiring only forward movement (e.g., running or jumping) without obstacles. In antmaze, if the agent flips upside down, it cannot recover, resulting in a zero reward for the episode. In locomotion tasks, the agent can still earn continuous rewards even with suboptimal actions, making its performance less sensitive to individual mistakes.
>
> Moreover, the reward structure in antmaze is binary—the agent either reaches the goal and receives a reward or fails and gets zero. This makes performance highly dependent on avoiding mistakes over long episodes (1000 steps), especially in larger mazes. For instance, an agent that quickly reaches the goal but fails at a corner near the end would still receive zero reward, whereas a slower agent that completes the maze in 1000 steps would receive full reward.
>
> This binary reward structure can feel counterintuitive. If I were the judge, I might award a higher score to the agent that reaches the goal faster, even if it fails near the end, as it demonstrates learning beyond simple imitation of expert behavior. So I suggest we should not take these grades too serious, not making it the only metric.
>
> As discussed in Appendix D.2, Table 11 shows that in smaller mazes, all methods perform similarly. However, in larger mazes, reverse-KL methods like GMPG and diffusion-QL degrade more significantly, while forward-KL methods like GMPO and QGPO remain more robust. This difference arises from the distinct characteristics of forward-KL and reverse-KL approaches. Forward-KL methods act as filters, selecting high-value actions from the existing policy. In contrast, reverse-KL methods encourage more radical exploration, sometimes pushing the policy toward unreliable or unsafe regions. This divergence is also visible in Figure 1, where forward-KL methods stay closer to the pretrained model, while reverse-KL methods deviate more rapidly.

---

> > ### Author Response · Authors · 2024-11-20
> >
> > ---
> >
> > > 'How do you calculate $\log{\pi}$ of the diffusion policies in GMPG? The introduction in lines 316-327 is too brief and the explanation in the appendix should be moved partially to the main text.'
> >
> > The calculation of the log probability $\log \pi(x)$ for diffusion policies in GMPG is based on the **instantaneous change of variables theorem**. This approach is detailed in the paper "FFJORD: Free-form Continuous Dynamics for Scalable Reversible Generative Models."
> >
> > In our case, sampling is governed by the differential equation:
> >
> > $\frac{dx}{dt}=v_{\theta}(x,t)$
> >
> > where $v_{\theta}(x, t)$ is the learned vector field. The log probability of the sample path is computed using the change in volume along the trajectory, as described by the log determinant of the Jacobian of the transformation. Specifically, the trace of the Jacobian of the vector field $v_{\theta}$ is used to account for the infinitesimal changes in volume during the sampling process. Mathematically, this can be expressed as:
> >
> > $\begin{bmatrix} x_0\\\ \log{p(x_0)}-\log{p(x_1)} \end{bmatrix} = \int_{t_1}^{t_0} \begin{bmatrix} v_{\theta}(\mathbf{x}(t), t) \\\ - \text{Tr} \left( \frac{\partial v_{\theta}}{\partial \mathbf{x}(t)} \right) \end{bmatrix} dt$
> >
> > where $x_1\sim\mathcal{N}(0,I)$.
> >
> > $\text{Tr} \left( \frac{\partial v_{\theta}}{\partial \mathbf{x}(t)} \right)$ is calculated using Hutchinson's trace estimator:
> >
> > $\text{Tr} \left( \frac{\partial v_{\theta}}{\partial \mathbf{x}(t)} \right) =  \mathbb{E}\_{p(\epsilon)}[\epsilon^T \frac{\partial v_{\theta}}{\partial \mathbf{x}(t)} \epsilon]$
> >
> > where $\epsilon$ is a random vector sampled from a standard Gaussian distribution.
> >
> > If the page limit allows, we plan to move this explanation from the appendix to the main text of the paper to make these details more accessible and clear for the reader.

---

### Author Response · Authors · 2024-11-24

Hello everyone,

Thank you for all your comments and suggestions!

We have added explanatory content to the revised paper to address some of the reviewers' questions. The relevant paragraphs and sections have been highlighted in blue for clarity.

The specific revisions can be found in:

- Section 2: Related Works
- Table 1
- Table 2
- Section 4.3
- Appendix A.2
- Appendix C.7.1

Additionally, to comply with the conference's page requirements, we adjusted the indentation of some formulas to mitigate page overflow caused by the new content.

---

### Meta-Review · Area_Chair_W3pu · 2024-12-21

**Metareview:**

This paper aims to simplify and standardize training generative policies in reinforcement learning (RL) through the introduction of GMPO and GMPG algorithms and a standardized experimental framework, GenerativeRL. While the authors provide a valuable unified perspective and modular framework, the submission lacks significant novelty, as GMPO and GMPG closely resemble existing methods like AWR with minimal innovation. Performance improvements on benchmarks like AntMaze are limited, and the theoretical connection between the proposed objectives remains underexplored. Additionally, presentation clarity and framework positioning as either a benchmark suite or a novel algorithm are ambiguous. Given these shortcomings, the paper falls short of meeting the standards for acceptance at ICLR.

**Additional Comments On Reviewer Discussion:**

During the rebuttal, reviewers raised concerns about the paper's lack of novelty, limited performance improvements in challenging tasks (e.g., AntMaze), and unclear positioning between proposing a framework versus new algorithms.

Reviewer *4kSb* noted that the forward/reverse KL analysis was well-known, while *FfDE* emphasized the importance of practical "tricks" ignored in the study. Reviewer *zQTz* pointed out errors in inference scheme descriptions and potential computational inefficiencies of GMPG. Authors addressed most issues by clarifying framework modularity, updating inference scheme descriptions, and adding theoretical connections in Appendix A.2.

However, critical concerns on novelty and underwhelming empirical results persisted, influencing the decision to reject.

---

### Decision · Program_Chairs · 2025-01-22

Reject